# A State-of-the-Art Review on CMOS Radio Frequency Power Amplifiers for Wireless Communication Systems

**DOI:** 10.3390/mi14081551

**Published:** 2023-08-01

**Authors:** Sofiyah Sal Hamid, Selvakumar Mariappan, Jagadheswaran Rajendran, Arvind Singh Rawat, Nuha A. Rhaffor, Narendra Kumar, Arokia Nathan, Binboga S. Yarman

**Affiliations:** 1Collaborative Microelectronics Design Excellence Centre (CEDEC), Universiti Sains Malaysia, Bayan Lepas 11900, Malaysia; sofiyah@usm.my (S.S.H.); jaga.rajendran@usm.my (J.R.); nuha@usm.my (N.A.R.); 2School of Computing, DIT University, Dehradun 248009, Uttarakhand, India; arvindsrawat@ieee.org; 3Department of Electrical Engineering, Faculty of Engineering, University of Malaya, Kuala Lumpur 50603, Malaysia; narendra.k@um.edu.my; 4Darwin College, Cambridge University, Cambridge CB3 9EU, UK; an299@cam.ac.uk; 5Department of Electrical and Electronics Engineering, Istanbul University, 34320 Istanbul, Turkey; sbyarman@gmail.com

**Keywords:** CMOS, radio frequency, power amplifier, wireless communication, efficiency, linearity, bandwidth

## Abstract

Wireless communication systems have undergone significant development in recent years, particularly with the transition from fourth generation (4G) to fifth generation (5G). As the number of wireless devices and mobile data usage increase, there is a growing need for enhancements and upgrades to the current wireless communication systems. CMOS transceivers are increasingly being explored to meet the requirements of the latest wireless communication protocols and applications while achieving the goal of system-on-chip (SoC). The radio frequency power amplifier (RFPA) in a CMOS transmitter plays a crucial role in amplifying RF signals and transmitting them from the antenna. This state-of-the-art review paper presents a concise discussion of the performance metrics that are important for designing a CMOS PA, followed by an overview of the trending research on CMOS PA techniques that focuses on efficiency, linearity, and bandwidth enhancement.

## 1. Introduction

Wireless communication systems based on radio frequency (RF) technology have become ubiquitous in electronic devices, particularly for mobile applications. As the number of subscribers continues to increase, the demand for RF communication systems also grows. Figure 1 illustrates the trend in mobile subscriptions by technology up to recent times [1]. The power amplifier (PA) block is a crucial component in transceiver design, as it consumes a significant amount of DC power. III-V compound semiconductors such as GaN HEMT, GaAs HBT, Si Bipolar, and SiGe HBT are commonly used to fabricate PAs, as they offer high linear output power and efficiency compared to the silicon CMOS process. However, these PAs are limited in their ability to integrate with CMOS baseband chips, hindering the development of a true single-chip radio or system-on-chip (SoC) [2]. To address this limitation, there was a recent focus on researching CMOS-based RF integrated circuits (RFICs) for wireless mobile applications [3,4,5,6,7,8,9,10]. However, CMOS-based PAs have several disadvantages due to their lower breakdown voltage, their higher substrate loss, and the unavailability of back via holes for the ground connection, which limit the achievable output power and efficiency compared to their III-V counterparts [11,12,13]. Despite their disadvantages, CMOS-based PAs offer a promising alternative due to their cost-effectiveness and potential for integration with other circuits. Several techniques were developed to improve the efficiency, linearity, and bandwidth performance of CMOS PAs [14,15,16,17]. 

Efficiency enhancement techniques are usually employed for highly linear PAs that consume more DC power. In contrast, linearity enhancement techniques are commonly employed for switching-based PAs, which have a low conduction angle that contributes to their non-linearity. Efficiency enhancement techniques target the efficiency of linear power amplifiers, while linearization techniques improve the linearity of highly efficient but non-linear power amplifiers [18]. As a result, there is an inevitable trade-off between efficiency and linearity performance in CMOS PAs. Recently, the bandwidth enhancement of CMOS PAs was also widely researched [19].

Wireless communication standards have rapidly expanded in recent years, with different applications utilizing various frequency bands such as L, S, C, UHF, and VHF, which are utilized by applications such as WLAN, Bluetooth, LTE, RFID, GSM, GPS, etc. [20,21,22]. To accommodate multiple standards in a single system, there is a growing need for CMOS PAs with enhanced bandwidth performance [23]. This state-of-the-art review paper provides an overview of the trending research efforts on the development of efficient and linear CMOS PAs with improved bandwidth performance for wireless mobile applications. This paper also discusses the limitations and challenges associated with CMOS PAs and outlines possible directions for future research.

## 2. Performance Metrics of CMOS Power Amplifier

Here, the terminologies and concepts associated with CMOS PAs’ performance such as output power, efficiency, stability, intermodulation distortion (IMD), adjacent channel power ratio (ACPR), error vector magnitude (EVM), and amplitude-to-amplitude (AM–AM) and amplitude-to-phase (AM–PM) modulations are presented.

### 2.1. Output Power

Achieving optimal output power in PAs is crucial, and it is a vital parameter that varies depending on the wireless application requirements. For example, to effectively transmit data, cellular applications require a linear output power of 28 dBm [24], while Bluetooth demands a significantly lower output power of 0 dBm [25]. Additionally, the distance of the signal transmission between the transmitter and receiver is directly proportional to the output power level, meaning that a higher output power level results in an extended transmission distance.

An example of this concept is observed in Bluetooth technology, where a Class 1 specification necessitates a linear output power of 20 dBm, resulting in a signal transmission distance of 100 m. In contrast, a Class 3 specification only requires an output power of 0 dBm, limiting the signal transmission distance to 1 m [26]. To determine the output power of the PA, (1) can be utilized with reference to Figure 2, assuming an output load resistance of R.
(1)$$P_{out}=\frac{{\left(\frac{V_{p-p}}{2}\right)}^{2}}{2R_{L}}$$

### 2.2. Efficiency

Efficiency is a crucial factor when it comes to assessing power consumption in PAs. Specifically, the drain efficiency and power added efficiency (PAE) are two commonly used parameters in the evaluation of PAs. In PAs, efficiency represents the amount of DC power that is converted into RF power. The PAE of a PA is a critical parameter that is typically used to assess the efficiency of PAs. In summary, drain efficiency and PAE play significant roles in evaluating the power consumption of PAs. The PAE of a PA is defined as [27]:(2)$$PAE=\frac{P_{o}-P_{in}}{P_{DC}}$$
(3)$$\frac{P_{o}}{P_{DC}}\left(1-\frac{1}{G}\right)=\eta_{D}\left(1-\frac{1}{G}\right)$$
(4)$$\eta_{D}=\frac{P_{o}}{P_{DC}}$$
where *P_o_* is the output signal power, *P_in_* is the input signal power, *P_DC_* is the DC power consumption, *G* is the gain of the PA, and *η_D_* is the drain efficiency of the PA. The drain efficiency and *PAE* are commonly close to each other when the gain of the PA is high. 

### 2.3. Stability

In PAs, stability is a crucial parameter, particularly for high-gain PAs. It is worth noting that PAs that are stable under 50 Ω load terminations may not be stable in realistic wireless environments due to antenna impedance mismatches that occur, such as those caused by nearby objects [28,29,30,31,32,33,34,35]. Therefore, achieving an unconditionally stable characteristic is necessary for PAs to remain stable despite load impedance mismatches [36]. The stability of a PA can be determined from Rollett (K), which has to be larger than unity. A PA is unconditionally stable if the value of *K* > 1. Meanwhile, if 0 < *K* < 1, it is considered conditionally stable, and, if *K* < 0, it is considered unstable. The *K*-factor is defined as
(5)$$K=\frac{1-{\left|S_{11}\right|}^{2}-{\left|S_{22}\right|}^{2}+{\left|∆\right|}^{2}}{2\cdot |S_{11}||S_{22}|}$$
where
$$\left|\Delta \right|=\left|S_{11}\cdot S_{22}-S_{21}\cdot S_{12}\right|$$

*S*_11_ is the input port voltage reflection coefficient, *S*_12_ is the reverse voltage gain, *S*_21_ is the forward voltage gain, and *S*_22_ is the output port voltage reflection coefficient.

### 2.4. Intermodulation Distortion

Due to the non-linearity that exists in the PA, it produces numerous extra terms at innumerable frequencies. The transistors utilized in the PA produce non-linearity due to the non-linear current and non-linear capacitance that rely on the device voltages [37,38]. The main non-linear elements that effect the PA’s performance are the transconductance (*g_m_*), drain-source capacitance (*C_ds_*), and gate-source capacitance (*C_gs_*) [39]. The Taylor series expansion can be used in order to determine the polynomial transfer function of non-linearity. The significant terms are the first, second, and third orders, which represent the gain, squaring, and cubing functions [40]. The PA’s non-linearity can be delineated by
(6)$$V_{o}(t)=a_{o}{+a}_{1}V_{i}(t){+a}_{2}{V_{i}}^{2}(t){+a}_{3}{V_{i}}^{3}(t)$$

Furthermore, the IMD that represents the linearity performance of a PA is determined by conducting a two-tone signals test in which approximately two spaced fundamental signal tones are supplied at the input of the PA. The amplitude of the two-tone signals is raised up to a certain level in which the third order products generate a signal with a higher amplitude above the noise floor [41,42,43]. The third order IMD (IMD3) product is the primary concern that needs to be dealt with because it is generated in-band of the fundamental signal frequency. The in-band-generated IMD product is difficult to eliminate using filters. Higher-order IMD products are small in amplitude to induce distortions and can be eliminated using filters [44]. The two-tone signal applied at the PA’s input is given as
(7)$$V_{i}(t)=\nu \;cos\;(\omega_{1}t)+\nu \;cos\;(\omega_{2}t)$$

Substituting (7) into (6) yields
(8)$$V_{o}\left(t\right)=a_{o}{+a}_{1}\nu \;\left[\cos\left(\omega_{1}t\right)+\cos\left(\omega_{2}t\right)\right]{+a}_{2}\;\nu^{2}{\left[\cos\left(\omega_{1}t\right)+\cos\left(\omega_{2}t\right)\right]}^{2}\;{+a}_{3}\nu^{3}{[\cos\;(\omega_{1}t)+\cos\;(\omega_{2}t)]\;}^{3}\;\;\;\;\;\;\;\;\;\;$$

From (8), the generated IMD components up to third order are shown in Table 1.

Thus, the amplitudes of both the upper and lower *IMD*3 products are defined as
(9)$$IMD3=\frac{3}{4}a_{3}\nu^{3}$$

Figure 3 illustrates the output power spectrum of a PA subjected to IMD. It can be observed that the upper and lower *IMD*3 products are generated in-band approximately to the fundamental tones at 2*ω*_2_ − *ω*_1_ and 2*ω*_1_ − *ω*_2_, respectively. 

### 2.5. Adjacent Channel Power Ratio

The adjacent channel power ratio (ACPR) is also a crucial metric that needs attention when a modulated signal is used as the input for the PA [45]. The ACPR values typically depend on the application of the wireless communication. Each wireless standard has adjacent channels located at different frequency offsets. For example, according to the 3GPP standard, the ACLR should be no more than −33 dB at the offsets of ±1.6 MHz and no more than −43 dB at the offsets of ±3.2 MHz. When a PA is operated with a modulated signal, it generates a bandwidth expansion that is commonly known as spectral regrowth. Spectral regrowth is a form of distortion since it results in adjacent channel interference [46,47]. Figure 4 is an exemplar of the output spectrum that illustrates the phenomena of spectral regrowth. 

The ACPR is defined in a frequency domain as the power ratio between the distortions in the sideband channels and the signal in the main channel. The ACPR is represented as follows:(10)$${ACPR}_{adj}=10\;log\;\frac{\int_{adj}S(f)df}{\int_{main}S(f)df}$$
where *S*(*f*) is the output signal’s power spectral density (PSD), and the term “*adj*” can be either the first lower or upper channel.

### 2.6. Error Vector Magnitude

Another vital metric to determine the linearity of a PA is the error vector magnitude (EVM). The EVM is a time domain parameter that is represented in terms of a constellation diagram. The rotation and compression of the constellation points are the primary contributor to the constellation distortion, as delineated in Figure 5. 

The error vector is expressed as the vector discrepancy of the measured and ideal constellation points in the I/Q plane. The *EVM* is a customary figure of merit (FoM) utilized to interpret the distortions inside the operating band that deteriorate the bit error ratio (BER). The *EVM* is given as
(11)$$EVM=\sqrt{\frac{{\Sigma_{m=1}^{M}|S_{ideal,m}-S_{measured,m}|}^{2}}{{\Sigma_{m=1}^{M}|S_{ideal,m}|}^{2}}}$$

### 2.7. Amplitude-to-Amplitude and Amplitude-to-Phase Modulations

The linearity of the PA can also be determined through the amplitude-to-amplitude (AM–AM) and amplitude-to-phase (AM–PM) modulations. When the PA is subjected to a large-amplitude input signal, the amplitude of the output signal tends to compress, and this decreases the linear output power that can be achieved. Moreover, the phase shift reflects the time delay in which, for an ideal case, it is undeviating at all amplitude levels. However, this is not the case in a real application, since the AM–PM modulation is not constant due to the existing non-linearity. A modulated input signal is expressed as [48]
(12)$$\varkappa \;(t)=A(t)\cdot \;cos\;[\;\omega_{c\;}t+\theta (t)]$$
where *A*(*t*) is the instantaneous amplitude, and *θ*(*t*) is the instantaneous phase.

AM–AM reflects the interrelation between the output signal’s amplitude and *A*(*t*). Meanwhile, AM–PM is the interrelation between the output signal’s phase deviation and *A*(*t*) [49]. Hence, the output signal is represented as
(13)$$y(t)=g\;(A(t))\cdot \;\;cos\;[\;\omega_{c\;}t+\theta (t)+\psi (A(t))]$$
where *g*(*A*(*t*)) and *ψ*(*A*(*t*)) are the AM–AM characteristics and AM–PM expression, respectively. The AM–AM and AM–PM characteristics are depicted in Figure 6 and Figure 7, respectively.

## 3. Efficiency Enhancement Techniques

Several techniques were researched and implemented as efforts to enhance the efficiency of CMOS PAs. Commonly utilized efficiency enhancement techniques include the on-chip transformer, Doherty, envelope tracking (ET), envelope elimination and restoration (EER), and out-phasing PAs.

### 3.1. On-Chip Transformer Power Amplifier

Numerous efforts were applied to utilizing the configurations of on-chip transformers by integrating them together with CMOS PAs. On-chip transformers have various convenient features such as impedance matching, DC isolation, ESD protection, differential-to-single-ended conversion, and single-ended-to-differential conversion. On-chip transformer structures can be classified according to how they are connected. The structures include series combining (voltage mode) and parallel combining (current mode) [50]. The aforementioned structures are delineated in Figure 8.

As the name implies, the series combination transformer has its secondary coils connected in series. The series implementation exhibits a voltage addition on the secondary side, which yields higher output power. This is due to the equal current of V_O_/R_L_ conducted by the winding of each transformer on the secondary side, as illustrated in Figure 8a [51]. The impedance observed by each amplifier is *n* times higher compared to the impedance seen when it is directly connected to R_L_. This is advantageous for the driver amplifier design and for the mitigation of the parasitic effects on the primary side. The transformer has to be highly symmetrical because mismatches could occur, which degrades the maximum efficiency and output power that can be achieved by the PA [52].

By utilizing a series combination transformer, a CMOS PA achieving watt-level output power was proposed by Aoki et al. (2002) [53]. The CMOS PA is employed in a single chip with integrated input and output matching networks. A distributed active transformer (DAT) utilizing a series combining structure is employed for power combining and impedance matching, enhancing the efficiency of the PA [54]. However, the implementation consumes a huge active area on the chip. Meanwhile, a CMOS PA with a smaller footprint was achieved by Javidan et al. (2010) by combining a transformer with different sizes inside each other [55]. The proposed transformer is designed with multiple primary single-turn inductors and a single secondary multi-turn inductor. 

The output of multiple PAs can be combined by using a series of combination 1:1 ratio transformers [56]. The output power of the PA can be varied by switching the individual transformers on or off [57]. A. Afsahi et al. (2010) utilized a transformer in order to parallelly combine signals to reduce losses on the secondary side as well as to achieve a better signal symmetry [58]. However, the number of turns in the secondary winding increased, which degrades the self-resonance frequency as well as consumes a huge chip area.

Moreover, K.H An et al. (2007) proposed a monolithic voltage-boosting parallel primary transformer that was utilized for power combining [59]. The transformer gave a voltage-boosting effect by increasing its turn ratio between the primary and secondary windings. An increased current was established in the secondary winding by interweaving multiple primary windings in parallel. An efficient power combining method was achieved by elevating both the secondary voltage and current. An interleaved transformer structure was proposed by A.R. Belabad et al. (2013), which achieved better efficiency compared to a 1:1 ratio transformer [60]. Although the interleaved configuration reduced the Q-factor and self-inductance, it had an enhanced coupling factor that is beneficial for transferring current via its magnetic field. The proposed transformer had two input ports and an output port. The drain terminals of different PAs were connected to the input ports of the transformer while the load was connected to its output port.

Furthermore, H. Ahn et al. (2017) proposed a dual-mode autotransformer-based parallel combining transformer (ABPCT) that provides high efficiency [61]. An individual PA was connected on the secondary side while multiple individual PAs were connected on the primary side. For high-output power operation, all the PAs were turned on. Contrarily, for low-output power operation, one or two discrete PAs were specifically turned on. The high- and low-power modes were selected by controlling the gate bias of each PAs. Autotransformers have lower power losses due to their reduced total series resistance, which is contributed to by the primary winding that is also a part of the secondary winding. An autotransformer-based PA attained an efficiency of 34% in low-power mode and 38% in high-power mode. 

In addition, J. Tsai et al. (2019) proposed a CMOS PA with a transformer-based two-stage dual-radial power splitting and combining mechanism [62]. Several advantages of this proposed transformer included in-phase signal splitting and combining, a compact network, the uniform distribution of DC supplies, and a symmetric supply and return path of the DC current. By utilizing the multi-layer structure of CMOS technology, the radial power splitting plane was implemented on a lower metal layer below the radial power combining plane. This contributed to a compact chip size since both the power splitting and combining shared the same chip area on top of each other. A shielding structure was employed to enhance the isolation between the input redial power splitting and output radial power combining networks. The maximum output power delivered by the PA was 29 to 31 dBm, with a maximum efficiency of 15–25%. J. Tsui (2019) also proposed a 5.3 GHz CMOS PA with a folded radial power splitting and binary power combining on-chip transformer [63]. The proposed architecture of the transformer provided an in-phase signal splitting and combining mechanism. The transformer was also designed with a compact size with uniform DC distribution as well as a symmetric DC current path. At the operating frequency of 5.3 GHz, the CMOS PA achieved a maximum output power of 31.3 dBm, with a peak PAE of 22%. 

Another on-chip transformer-based CMOS PA was presented by H. Choi et al. (2020), in which a 1:1 turn spiral transformer balun was implemented with a stacked main PA configuration in order to sustain a high voltage supply [64]. The main PA was constructed in a differential push–pull configuration in order to attain high output power and stability. The 1:1 transformer was employed with an equal coupling line length and an equal inductance in order to achieve an optimum impedance with minimum losses. The proposed CMOS PA also utilized an anti-phase biasing technique to achieve a higher linear output power. The CMOS PA achieved a maximum output power of 29.9 to 30.3 dBm, with 40% to 45% across its operating bandwidth of 0.82 to 1 GHz. 

T. Wang et al. (2021) presented an on-chip transformer-based digital CMOS PA in which the transformer acted as a balance-compensated matching network [65]. A series-loaded compensation capacitor was utilized to enhance the balance response of the transformer, which improved the efficiency of the CMOS PA. The transformer was implemented in a four-way series combining configuration. The series combining configuration aided in reducing the voltage stress on the transistors and the impedance transformation ratio of the matching network. The designed transformer was capable of simultaneously providing four-way power combining, balance compensation, wideband impedance matching, and differential-to-single-ended conversion. The proposed CMOS PA delivered a maximum output power of 32.4 dBm at 2 GHz. The maximum drain efficiency achieved was 53.8% at 1.8 GHz. 

B. Yang et al. (2021) presented a CMOS PA with a reconfigurable self-coupling cancelling transformer [66]. The proposed transformer had an improved tuning range of the turn ratio that was utilized to boost the PA’s efficiency at deep back-off output power. The core of the PA was constructed in a quadrature digital PA configuration with I-Q cell sharing and a hybrid Doherty architecture. The PA also employed a switched/floated capacitor for impedance tuning. The CMOS PA achieved a maximum output power of 30.3 dBm, with 36.6%, the highest system efficiency, at 2.4 GHz. B. Yang et al. (2023) also proposed a watt-level CMOS digital PA using a reconfigurable power-combining transformer [67]. The transformer as employed in a voltage-mode power combining configuration in order to attain high output power across a wide frequency bandwidth. The output power of the CMOS PA at the fundamental signal was further enhanced by using a LC circuit that suppresses the harmonics. The reconfigurable output matching network was constructed using the power-combining transformer together with four switched capacitors at the inputs and a fixed capacitor and output. Additionally, the dynamic range of the digital PA was enhanced by mitigating the LO leakage by using a 12-bit power digital-to-analog converter (power DAC). The CMOS PA delivered a maximum output power of 32.67 dBm, with a peak PAE of 35.5% at 2 GHz.

### 3.2. Doherty Power Amplifier

The Doherty PA (DPA) was initially proposed by W. H. Doherty, comprising a carrier and a peaking amplifier [68]. A typical block diagram of a classic DPA is depicted in Figure 9 [69]. The back-off efficiency of the PA is improved through its load impedance variation. This is realized by implementing a current source at the output terminal. Figure 10 illustrates the concept of the DPA. 

Based on Figure 10, the auxiliary amplifier is denoted by “Aux”, and the main amplifier is denoted by “Main” [70]. The auxiliary amplifier is commonly operated in class C mode while the main amplifier is operated in class AB mode. When the auxiliary amplifier is inactive, the main amplifier faces a load resistance of R_L_. Whereas, if the auxiliary amplifier is active and generates a current, *I*_aux_, the main amplifier, sees a load impedance as
(14)$$Z_{main}=R_{LOAD}\left(1+\frac{I_{aux}}{I_{main}}\right)$$

From (14), it is evident that the load impedance of the main amplifier can be varied by controlling the source current of the auxiliary amplifier (*I*_aux_). This feature of the DPA is utilized to enhance its back-off efficiency, as illustrated in Figure 11 [71]. A constant efficiency across output power was attained by utilizing the DPA technique in CMOS, as presented by N. Deltimple et al. (2015) [72]. S. Hu et al. (2016) proposed a class G DPA that enhances the efficiency at deep back-off output power [73]. A hybrid combo of class G and Doherty improved the deep-back off efficiency without extra complexity at the input and output passive networks. 

E. Kaymaksut et al. (2015) presented a CMOS dual-mode Doherty PA with a four-way symmetrical hybrid transformer [74]. The transformer mitigated the impedance imbalance that exists in the PA. The PA employs a switching mode technique with the combination of the Doherty technique, which realizes the high-power and low-power modes. The PA also utilized an adaptive bias circuit that further enhances its linearity and back-off efficiency across its frequency of operation. The proposed transformer utilized the combination of a distributed active transformer (DAT) and a figure-eight-shaped transformer [75,76]. The designed transformer was a 2 × 2 hybrid transformer that utilizes the combination of two DATs with a figure-eight-shaped transformer. In LP mode, only two sub-PAs are switched on, while in HP mode, all the PAs are made active. The PA operates from 1.7 to 2.1 GHz, with a maximum output power of 26–28.2 dBm, as well as a peak efficiency of 25.5–34%. The achieved linear output power was 21–23.4 dBm, with a corresponding efficiency of 16–23.4%. 

Further, C.S. Levy et al. (2016) presented a series Doherty PA (SDPA) that is capable of delivering a higher output power than a conventional DPA [77]. In an SDPA, the load voltage is two times higher compared to the voltage of each amplifier separately because of the parallel load impedance of the peaking amplifier. To achieve an efficiency approximate to that of class B operation, the main amplifier is operated in deep class AB operation. In addition, S. Traiche et al. (2017) presented a highly efficient fully integrated CMOS DPA [78]. The proposed DPA comprised a main and an auxiliary amplifier that are operated in class AB and class B, respectively. Both amplifiers utilized the cascode structure to endure the high supply voltage and to achieve high input–output isolation. A Wilkinson divider was used at the input for signal splitting. As for the transmission line, a π-network-based dephasing circuit and impedance inverter were utilized, as well as resonant tanks, to terminate the harmonics. 

In addition, D. Jung et al. (2019) presented a Doherty CMOS PA with multi-gated transistors (MGTRs) in the main PA core [79]. This Doherty CMOS PA also employed a class C mode auxiliary PA and a synthesized-transformer-based parallel output network with a second-harmonic control circuit. Each MGTR consisted of four transistors with different sizes and was biased to attain a near-zero third-order transconductance operation. The MGTR and the second-harmonic control circuits both aid in suppressing the third-order intermodulation distortion of the proposed Doherty PA. The MGTR Doherty CMOS PA achieved a maximum output power of 27.2 dBm, with 24.5% PAE at 5.8 GHz.

Y. Yin et al. (2020) proposed a CMOS Doherty PA using digital implementation with a parallel combining transformer for deep back-off efficiency enhancement [80]. The parallel combining transformer offers a dynamic load modulation for the digital Doherty PA in an ultra-compact size. The designed digital Doherty PA consisted of four sub-PAs that are controlled in a precise manner. The load modulation was performed by connecting each of the differential sub-PA pairs to the terminals of the two adjacent primary coils of the power combiner. The digital Doherty PA delivered a maximum output power of 21.4 dBm, with a peak PAE of 31.3% at 1.5 GHz. 

### 3.3. Envelope Tracking Power Amplifier

In an envelope tracking power amplifier (ETPA), a supply modulation mechanism is employed to manage the supply voltage of the PA with the corresponding RF input voltage while maintaining a fixed load resistance [81]. This contributes to the efficiency improvement at the low-output power region by minimizing the power dissipation for systems with a high PAPR [82]. The ET supply modulator tunes the supply voltage level of the PA with respect to the time-varying envelope of the RF input signal in order to keep the PA in stable compression mode. This contributes to the efficiency enhancement of the PA. Figure 12 illustrates the voltage difference between the supply voltage and the RF output signal under a modulated signal [83].

An envelope amplifier is utilized for biasing the RFPA according to the input envelope RF signal, as depicted in Figure 13. An envelope detector on the input path in employed to retrieve the envelope information. The envelope information can also be retrieved by utilizing digital baseband processing [84].

Figure 14 depicts the block diagram of an ETPA together with its relative waveforms. The ET method is commonly integrated with a linear power amplifier (LPA), and the RF signal amplified contains both the amplitude and phase information. The power supply in an ETPA is mainly focused on enhancing the efficiency of the PA, and its overall efficiency includes both the efficiency of the PA and the envelope amplifier.

J. L. Woo et al. (2014) introduced a dynamic stacked CMOS ETPA that achieved high efficiency during low-output power operation [85]. The proposed ETPA achieved high efficiency by utilizing a 3.4 V maximum voltage. Furthermore, B. Park et al. (2016) proposed an ETPA with an improved ET supply modulator [86]. By integrating this supply modulator, an enhanced efficiency was achieved at all output power levels. There were also second-order harmonic control mechanisms employed at the input and output of the ETPA that further improved its efficiency. Moreover, an ETPA with a dual-mode supply modulator was presented by Ham et al. (2016) [87]. A dual-mode supply modulator was utilized for envelope tracking and average power tracking. This modulator was designed based on a hybrid buck converter comprising switching and wideband linear amplifiers. This ETPA achieved a peak efficiency of 45.4% under the envelope tracking mode. 

### 3.4. Envelope Elimination and Restoration Power Amplifier

The envelope elimination and restoration (EER) technique was introduced by L. R. Khan in 1952 [88]. Contrary to ETPA, the EER technique is implemented in a non-linear PA (NLPA). EER adopts a supply modulation method in which it splits the RF modulated signal into envelope and phase signals and later combines it at the output of the NLPA. A high-speed supply modulator is employed for the supply voltage modulation of the NLPA. Figure 15 delineates the block diagram of an EERPA together with its key waveforms. 

Referring to Figure 15, the RF input signal that consists of amplitude and phase information is handled by two different paths, which are the envelope and phase paths. At the envelope path, the input signal’s envelope is identified by the envelope detector and is utilized as the reference voltage for the supply modulator. The supply modulator then generates an output voltage that follows the input signal’s envelope and supplies the voltage to the NLPA. In contrast, at the phase path, the limiter is utilized to eliminate the input signal’s envelope information and generate a constant amplitude signal with only the phase information. The amplitude information is restored by the supply voltage given by the supply modulator. The delay unit is utilized to synchronize both the envelope and phase paths. 

A. Mamdouh et al. (2020) presented a fast transient supply modulator for an EERPA [89]. The efficiency was improved at a high frequency due to the fast transient response of the supply modulator compared to conventional hybrid mode supply modulators. K. Oishi et al. (2014) proposed an EERPA with an envelope/phase generator based on a mixer and a limiter that generates a wide-bandwidth phase signal [90]. A delay locked loop-based timing aligner was implemented with a variable high-pass filter to compensate for the timing mismatch. Furthermore, W. Yuan et al. (2015) explored a digital-based implementation and proposed an EERPA with a class E configuration and a digitally controlled current DAC modulator [91]. A switched capacitor-based DAC was presented to control an open-loop transconductor that serves as a current modulator. The current modulator modulated the amplitude of the current supplied to the class E configured PA. 

X. Liu et al. (2019) proposed an EER envelope-shaping PA with an AC-coupling supply modulator [92]. The supply modulator comprised a 25 MHz three-level switching amplifier and a wideband-assisting linear amplifier. The AC-coupling method utilized the shaped envelope to mitigate the power consumption of the linear amplifier and generate a higher voltage than its supply. A high-efficiency, small-output voltage ripple and a fast transient response were achieved by implementing a switching amplifier with three-level real-time flying capacitor calibration. A multi-loop control was utilized in order to further increase the switching amplifier speed and, thus, enhance the overall efficiency. As for the PA, a mode-switching PA was implemented in a differential cascode structure, and bond wires were utilized as its DC-feed inductor. To improve its efficiency, the common-gate voltage of the cascode PA was dynamically biased with respect to the instantaneous supply voltage. A capacitive-charging-acceleration method was also implemented to further increase the efficiency [93]. The supply modulator achieved an efficiency of 88.7%, while the overall EERPA achieved an efficiency of 35.7%. 

### 3.5. Out-Phasing Power Amplifier

An out-phasing PA utilizes the method of separately splitting and amplifying the signals then combining them at the output. The signal with amplitude modulation is given as the vector sum of two constant envelope phase-modulated signals. An improved efficiency is achieved by modulating the differential phase while keeping the amplitude of the input signal constant. For out-phasing techniques utilizing non-linear components, amplitude modulation is conducted by combining the outputs of two PAs THAT are supplied with constant envelope phase-modulated signals [94,95,96,97,98]. The architecture of the out-phasing technique is shown in Figure 16, and its principle of operation is depicted in Figure 17.

A modulated signal can be represented, as shown in (15), and its amplitude and phase can be split into two constant amplitude and phase-modulated signals (*S*_1_(*t*) and *S*_2_(*t*)), as shown in (16) and (17).
(15)$$S_{out}\left(t\right)=A\left(t\right)cos(\omega_{c}t+\emptyset \left(t\right))$$
(16)$$S_{1}(t)=\frac{A_{max}}{2}cos(\omega_{c}t+\emptyset (t)+\theta (t))$$
(17)$$S_{2}(t)=\frac{A_{max}}{2}cos(\omega_{c}t+\emptyset (t)-\theta (t))$$
where *S_out_*(*t*) *= S*_1_(*t*) *+ S*_2_(*t*), *A*(*t*) is the time-dependent amplitude, and *ϕ(t)* is the time-dependent phase of the original signal, *S*(*t*); *ω_c_* is the angular frequency of the carrier, and *A_max_* is the maximum value of *A*(*t*). The angle *θ*(*t*) is defined as
(18)$$\theta (t)={cos}^{-1}\frac{A(t)}{A_{max}}$$

Since *S*_1_(*t*) and *S*_2_(*t*) are constant envelope phase-modulated signals, the signal amplification can be conducted by utilizing a non-linear PA. The outputs of the PA can be joined by using a combiner circuit in order to obtain *S_out_*(*t*). 

Z. Hu et al. (2016) proposed an RFDAC-based out-phasing PA with an integrated combiner [99]. Increased efficiency was achieved by utilizing a dynamic amplitude control in a quasi-load insensitive class E configured PA. The amplitude was controlled through a digitally divided method for the branch amplifier. Highly reactive and severe load modulation conditions were avoided through this method, which contributed to the minimization of power losses. 

Furthermore, a CMOS class D configured multi-level out-phasing transmitter with on/off control logic method was presented by M. Martelius et al. (2016). The PA consisted of eight individual amplifiers with cascoded output stages [100]. The PAs were selectively activated in pairs to achieve different amplitude levels as a means of improving the back-off efficiency. A. Banerjee et al. (2017) proposed a CMOS class E configured out-phasing PA with a novel passive combining circuit [101]. The designed passive combining circuit contributed to achieving enhanced output power and efficiency. The combiner comprised a power enhancement circuit and an efficiency enhancement circuit integrated together. Moreover, A. Ghahremani et al. (2018) presented a class E out-phasing PA by conducting load-pull analyses [102]. These load-pull analyses were carried out to rotate and shift the power and efficiency contours. This method was utilized to improve the efficiency at deep back-off output power. The rotation of the power and efficiency contours were realized by tuning the LC tank’s resonance frequency and duty cycle scaling factor. This class E PA consisted of transistors that serve as a square-wave input-signal-driven switch with a duty cycle of 50%. Parallel bond-wires with a high Q-factor were utilized as the DC-feed inductors for each PA. The cascade inverters employed in the PA were used as a driver for the switch. The duty cycle of the switch was controlled by using the off-chip control voltage. The switch capacitor banks were used to vary the resonance frequency of the LC tank at the switching nodes. 

M. Martelius et al. (2020) proposed an out-phasing CMOS PA with the combination of a polar mechanism that implements a tri-phasing modulation [103]. The combination of the polar and out-phasing components mitigated the linearity-degrading effects while maintaining the back-off efficiency. The CMOS PA comprised eight class D units on a single chip. The power combiner of this CMOS PA was implemented using Marchand baluns that consist of input transmission lines and coupled-line architecture on a printed circuit board. This CMOS PA achieved a maximum output power of 29.7 dBm, with a peak PAE of 34.7% at 1.7 GHz. 

A. Banerjee et al. (2020) presented a multi-mode class E out-phasing CMOS PA with an integrated passive combining circuit [104]. Multiple class E PAs were turned ON and OFF in order to configure the efficiency of the PA at low power levels. This out-phasing CMOS PA also utilized an efficiency enhancement circuit (EEC) which further assisted in improving the efficiency at back-off output power. In an out-phasing PA, the in-phase components are added together and increase the load power, while the out-of-phase components cancel each other out and reduce the load power. In order to enhance the back-off efficiency, the power loss due to the out-of-phase components should be minimized without trading off the in-phase operation. Thus, an EEC is employed for this purpose. The proposed EEC is realized using an inductor that is connected between the drains of the main PAs. Table 2 summarizes the recent state-of-the-art CMOS PAs with efficiency enhancement techniques.

Figure 18 delineates the plot of the PAE achieved by the aforementioned techniques across the achieved output power. The proposed techniques are capable of delivering a maximum output power of more than 20 dBm. It can be observed from Figure 18 that the on-chip transformer and the out-phasing techniques are capable of delivering watt-level maximum output power for CMOS PAs. This is due to the usage of power-combining architectures in these techniques that boost the maximum output power. The PAE achieved by these techniques was mostly more than 30%. The highest PAE was achieved by using the EER technique; however, the research was limited to only simulated data. 

## 4. Linearity Enhancement Techniques

Non-linearity in PAs is mainly contributed to by signal distortions. In CMOS MOSFETs, the distortions are generated by the inherent substrate non-linearity, trans-conductance, and parasitic capacitances, mainly the gate-source capacitance (C_gs_) [105,106,107]. AM–AM and AM–PM are some of the distortions that are caused by parasitic capacitances. The AM–AM distortion is minimal at a low frequency, and it increases as the frequency increases due to the non-linear C_gs_ [108]. 

AM–PM is mainly impacted by the MOSFET’s transconductance, which can be mitigated with suitable input matching network phase compensation techniques [109]. Intermodulation distortion (IMD), especially the third-order IMD, is another type of distortion that impacts the linearity performance of PAs. In a two-tone test, the third-order IMD occurred close to the fundamental tones and was arduous to filter out. The third-order IMD increased by 3 dB for every 1 dB increase in the fundamental tones [110]. In addition, spectral regrowth is another form of distortion in the adjacent frequency channels in which the power regrows. These adjacent channels might be assigned for different communications, which generates perturbations [111].

High linearity is necessary to fulfill the stringent protocols. Hence, linearization techniques are utilized to mitigate the distortions inside the operating bandwidth and to reduce the interferences from the adjacent frequency bands. Numerous techniques were employed to enhance the linearity of CMOS PAs, which include feedback techniques such as Cartesian loop feedback, polar loop feedback, feedforward, linear amplification using non-linear components (LINC), analog pre-distortion (APD), digital pre-distortion (DPD), and adaptive biasing.

### 4.1. Feedback Technique

The feedback method is conducted to improve linearity by connecting a fragment of the output signal back to the input of the PA [112]. The most common feedback techniques are the Cartesian and polar feedbacks. The feedback technique is advantageous due to its simplicity. However, its input and output signals need to be simultaneously operated, which is impossible to be realized due to the delays in the circuits. This serves as the major bottleneck of the feedback method. Figure 19 delineates the feedback principle in which part of the output signal (feedback signal) is connected back into the PA’s input after passing through a delay compensation mechanism, β.

#### 4.1.1. Cartesian Feedback Technique

Cartesian feedback is a technique that encompasses the linearization of the overall transmitter system. In the Cartesian feedback method, the baseband I and Q signals are upconverted to the carrier frequency, and the signals are then amplified. The signals are later sampled and downconverted back into the quadrature component, which is connected back to the input of the transmitter. 

Error amplifiers are employed to compare the signal to the original baseband input signals [113]. The delay that exists in the feedback loop limits the bandwidth of the transmitter. Figure 20 depicts a simple block diagram of a Cartesian feedback implemented in a PA [114].

L. Tee et al. (2006) presented a Cartesian feedback implemented in a CMOS PA for EDGE application [115]. The PA comprised a pre-driver, a driver, and a main amplifier. The main amplifier was operated in class C, which attained high efficiency. A class AB transistor was integrated with the class C amplifier to achieve stability at all input levels at the output stage. Furthermore, a Cartesian feedback linearization for CMOS Zigbee PA was proposed by A. Atress et al. (2016) [116]. A class E operation was adopted to obtain a high efficiency. The PA was tested with π/4 DQPSK modulation at 868 MHz in order to verify its linearity performance. A resistor was used at the input matching network to improve the stability of the PA. An inductor and a capacitor connected in series in the output matching network were utilized for waveform shaping. The employed Cartesian feedback system was capable of lowering the spectral regrowth at the adjacent channels.

#### 4.1.2. Polar Feedback Technique

Unlike a Cartesian feedback, which feedbacks the I and Q data, the polar technique feedbacks the amplitude and phase into the input of the system. The difficulty with this approach is that the feedback bandwidths of the amplitude and phase components are different from each other. This deteriorates the available loop gain to either the amplitude or phase path because one path needs a feedback bandwidth that limits the loop gain, while the other path requires a larger loop gain. Figure 21 shows the block diagram of a polar feedback system that addresses both amplitude and phase distortion with separated loops [117].

In the polar feedback technique, a local oscillator and a mixer are employed to attenuate and downconvert the output signal of the PA. The downconverted output signal is compared with the envelope of the incoming intermediate frequency (IF). Moreover, the error amplifiers in the negative feedback loop are employed to control the bias condition of the PA. 

A polar transmitter with a current-mode class D CMOS PA was presented by T. Nakatani et al. (2013). This polar transmitter also consisted of a digital pulse width modulation (DPWM) algorithm that was employed to mitigate spurious signals linked with the digital input signal envelope [118]. DPMW was mainly employed to minimize the noises induced by the clock via a minimum quantization error dithering (MQED) method. This approach spreads out clock spurs, reduces quantization errors, and mitigates peak spurs. 

Further, S. Zheng et al. (2013) presented a digital polar transmitter comprising a six-bit CMOS PA array [119]. The proposed design employed a highly linear charge-mode switched-capacitor digital polar modulator (DPM) in place of the conventional switched-current DPM. A cascode common-source configuration in the differential structure was utilized in the PA. The replica of the main PA was integrated with a comparator and a resistor. The replica was employed to regulate the main PA’s bias level with an analog feedback loop in which it mitigated the AM path’s distortion. 

### 4.2. Feed-Forward Technique

Feed-forward linearization is employed to mitigate distortions at the output of the PA. The feed-forward technique focuses on the subtraction of the harmonics and intermodulation components from the output spectrum of the PA [120]. Feed-forward comprises two cancellation loops based on two amplifiers: the main amplifier and the error amplifier. Figure 22 shows the block diagram of a feed-forward-based PA. The input signal is split into two, in which one portion is amplified via the main PA, and another portion is processed via the delay element. The feed-forward method is advantageous for wideband linearization, but it increases the implementation cost and circuit complexity. 

The feed-forward approach was realized in a 24 GHz cascode CMOS PA, as demonstrated by Y. H. Chen et al. (2015) [121]. The PA employed a successive second-order intermodulation (IMD2) feed-forward cancellation. The main path comprised a cascode PA operated in class AB, while the auxiliary path comprised successive IMD2-generating common-source amplifiers. The transistors in the common-source amplifiers were operated in the class B region, in which it detected the fundamental signals and generated the maximum IMD2 components and moderate fundamental components. Further, the transistor in the final stage generated IMD3 components that were utilized to cancel the main path’s IMD3 components. A diode-connector-based level shifter was employed at the final stage in order to realize the biasing voltage in class B and also to allow the very-low-frequency signals to pass through. The PA was capable of effectively mitigating the IMD3 by reducing the third-order transconductance, which is the major contributor to IMD3. The IMD3 value was enhanced by 20 dB at the “sweet spot” located at an output power of 5 dBm, with the auxiliary stage being active. The IMD3 measurement was carried out with a 1 MHz spacing two-tone test at 24 GHz, while the ACPR measurement was completed with 64 QAM, at a 1 MHz symbol rate.

### 4.3. Linear Amplification Using Non-Linear Components Technique

The linear amplification using non-linear component (LINC) is usually employed with highly efficient PAs such as class C, D, E, or F. In LINC, the modulated input signal is split into constant envelope and phase-modulated signals. These signals are independently amplified by two PAs with the same features. The two amplified signals are later combined together at the output with fewer distortions [122]. The separated signals are constant envelope, thus they are insensitive to the non-linearity of the PAs. Therefore, the employment of non-linear switching-mode PAs is advantageous when implemented in LINC techniques. Figure 23 illustrates the typical block diagram of a LINC technique.

H. Lee et al. (2010) presented a multi-mode LINC PA that was employed with a differential two-pair class AB CMOS PA [123]. The PA comprised two pairs of driver stages and main power stages that employed a cascode structure and stabilizing feedback circuits. The multi-mode LINC PA was capable of enhancing its linear output power by 7% and its efficiency by 8% compared to the conventional method. H. Lee et al. (2013) also presented a hybrid polar-LINC PA that employed a combination of polar and LINC mechanisms [124]. This hybrid polar-LINC PA was configured in a differential structure comprising two pairs of driver stages and power stages and an output combiner. The driver and power stages were both based in class E operation. The proposed power combiner comprised a transmission line transformer (TLT) with isolation resistors. The employed isolated TLT avoided distortion induced by the mutual load-pulling effects, which occur in non-isolated combiners [125]. The isolated combiner allowed a more linear and precise signal combination that resulted in ACLRs of −36 dBc and −47 dBc under 5 MHz and 10 MHz WCDMA signals, respectively.

### 4.4. Pre-Distortion Technique

Pre-distortion employs an approach of intentionally introducing distortions in terms of amplitude and phase that are in contrast to a PA’s distortion [126]. This results in the cancellation of distortions and, thus, enhances the linearity. The pre-distortion technique provides a wide-bandwidth operation due to its open-loop feature, which makes it advantageous over feedback techniques [127]. Pre-distortion is also advantageous over the feed-forward technique due to its simplicity and low power consumption [128]. The pre-distorter mechanism is implemented at the input of the PA, as shown in Figure 24. Pre-distortion is highly flexible, effortlessly reconfigurable, and suitable for wideband application. However, the PA’s characteristics’ variation is subjected to component tolerances. The pre-distortion mechanisms can be employed in an analog or digital manner. 

#### 4.4.1. Analog Pre-Distortion Technique

Analog pre-distortion (APD) employs a non-linear mode device to pre-distort the input signal prior the PA. RF pre-distortion is usually implemented in an analog mechanism due to the complexity of the sampling frequency needed to be digitally processed. 

K. Y. Son et al. (2012) proposed a CMOS PA with an integrated digitally controlled APD [129]. This APD had two external digital control words, which are the I-path and Q-path control words. These digital control words were utilized to control the AM–AM and AM–PM distortions in the PA. The driver stage of the PA was employed in a vector modulator manner, so the gain and phase can be varied via the vector sum method [130,131]. The linear output power of the PA was enhanced by 6.5% with this approach. Furthermore, a 15 GHz DPA with an integrated APD as presented by N. Rostomyan et al. (2018) [132]. The main and peaking paths were both constructed with two-stage amplifiers. In the input and output power combiners, high-pass π-networks were utilized due to their advantages in terms of stability and a compact area compared to a λ/4 transmission line. The APD in this design focused on the gain non-linearity issue of a symmetrical DPA. The linear output power and gain flatness were achieved by controlling the supply and bias voltages of the envelope detector employed in the PA. An improvement of 9.1% in the linear output power was observed via the implementation of the proposed APD.

Furthermore, S. Mariappan et al. (2021) presented a CMOS PA with a wideband pre-distortion (WPD) mechanism that consisted of a hybrid feedback with both active and passive networks [133]. The WPD provided both gain and phase cancellation schemes that aid in mitigating the third-order intermodulation product. The WPD exhibited a capacitive response that is in opposition to the main PA’s inductive response. An active load was implemented at the WPD instead of an RF choke, which generated the capacitive response. The proposed CMOS PA with WPD achieved a maximum output power of 24 dBm, with a peak PAE of 35.5% at 2.45 GHz. The achieved linear output power was 20 dBm across the operating frequency bandwidth of 0.8–3.3 GHz. S. Mariappan et al. (2021) also presented a CMOS PA with a broadband pre-distortion (BPD) that used the same aforementioned mechanism [134]. However, this CMOS PA as implemented on a Roger’s RO4000/FR4 PCB, which achieved a maximum linear output power of 21 dBm across the frequency bandwidth of 0.4–2.8 GHz. The peak efficiency achieved is more than 35%. 

S. Mariappan et al. (2021) proposed another CMOS PA with an integrated digitally assisted analog pre-distorter (DAAPD), which offers a reconfigurable linearization scheme [135]. The DAAPD was utilized to tune the interstage load impedance between the driver stage and the main stage. The load impedance was reconfigured by varying the transconductance of the driver amplifier and its respective active load. The DAAPD consisted of a digital bias controller that comprised a seven-stage voltage generator with a band-gap reference voltage, buffer, and op-amp, which was employed to provide the biasing voltage that controls the transconductance. The DAAPD was also employed to optimize the process–voltage–temperature variation in the CMOS PA that contributes to its robustness. The DAAPD generated a capacitive response as opposed to the inductive response of the main PA that enhances the linearity. This CMOS PA exhibited a maximum output power of 27–28 dBm across the frequency bandwidth of 1.7–2.7 GHz. The linear output power achieved was 24–25 dBm.

Based on the aforementioned design, S. Mariappan et al. (2023) further improved the presented design by proposing a reconfigurable CMOS PA with digital linearizers and a tunable-output impedance matching network for PVT robustness [136]. The digital linearizer was implemented at the driver stage of the CMOS PA, which acted as the wideband pre-distorter. The tunable-output matching network was realized via an interleaved planar transformer with a switched capacitor at both the primary and secondary windings. The switched capacitor was capable of maintaining the efficiency and the output power of the CMOS PA across its operating frequency bandwidth as well as PVT, while the digital linearizer conducted the linearization tuning. The proposed mechanisms for the CMOS PA achieved a linear output power of 24–25 dBm and a PAE of 34.5%–38.8% across the frequency of 1.7–2.7 GHz. 

In addition, J. Tsai (2023) proposed a CMOS PA with a high-power APD constructed with a cascode cold-FET [137]. The CMOS PA was comprised of three differential main PA units, a driver amplifier, a two-stage dual-radial power splitter/combiner and a pre-distorter as the linearizer. The cascode cold-FET had a higher breakdown voltage that can sustain a higher input power and generated a larger gain expansion region compared to conventional common-source cold-FET implementation. This allowed the APD to conduct linearization at a higher-output power region and further enhanced the linear output power of the proposed watt-level CMOS PA. The proposed CMOS PA operated at 5.3 GHz, with a maximum output power of 30.1 dBm and a peak PAE of 18%. The achieved linear output power was 21 dBm after linearization was conducted. 

#### 4.4.2. Digital Pre-Distortion Technique

The digital pre-distortion (DPD) technique is gaining popularity due to the rapid development of digital signal processing (DSP) [138]. Due to the limitations in terms of complexity, cost, and efficiency, the DSP-based DPD is restricted to less than a 100 MHz operation bandwidth. The DPD mechanism is commonly implemented with direct and indirect learning architectures [139,140,141]. 

Y. Cho et al. (2014) presented a CMOS DPD PA with an off-chip open-loop DPD algorithm that only uses a single look-up-table (LUT) [142]. The DPD was employed for a large dynamic range of output power via a LUT-based average output power control. An enhancement of 8 dB was achieved in ACPR by utilizing the proposed DPD. 

Moreover, a two-core digital CMOS PA with on-chip integrated DPD was presented by A. Wong et al. (2017) [143]. A polar architecture and an on-chip 1-D LUT were employed in the proposed design. The digital PA adopted a double cascode configuration. The NAND gates were employed to drive the device’s transconductance with inputs of amplitude and phase switching at the rate of the carrier frequency. The amplitude data were obtained via the on-chip DSP in four 12-bit data streams at a rate of one-fourth of the carrier frequency. On the other hand, the phase data were obtained via two sets of four parallel 10-bit data streams that were converted into two series of 10-bit data streams before they were supplied into the I-Q DAC. A piecewise linear DPD was employed in this design, which achieved linear output power of 26.8 dBm and 25 dBm for the 20 MHz 802.11g and 40 MHz 802.11n signals, respectively.

### 4.5. Adaptive Biasing Technique

Linearization can also be conducted by controlling the transconductance of the CMOS PA. Since this tranconductance is directly dependent on the gate voltage of the transistor, adaptive biassing circuits are extremely beneficial for this implementation. They only provide a variable bias current when an input signal is present. In the absence of such input, they provide a constant, precisely controllable quiescent current. In this manner, the quiescent power consumption is decreased without compromising the transient performance. 

W. Kim et al. (2014) proposed a CMOS PA with an adaptive biasing technique for EDGE/GSM application [144]. To improve the AM–AM characteristics, this CMOS PA was adaptively biased according to its input power level. The adaptive bias circuit was designed to enhance bias voltages when the input signals were large, thereby increasing the 1 dB output power in proximity to the peak power. Also, the non-linear gate–drain capacitance (C_gd_) of the power transistors, which is a primary source of AM–PM non-linearity, was effectively linearized by adding a series capacitor. The proposed CMOS PA achieved PAEs of 22% and 23% at linear output powers of 28.5 dBm and 27.5 dBm, at 870MHz and 1.8 GHz, respectively. 

H. Kim et al. (2015) introduced a CMOS PA with the implementation of a dynamic bias switching method via digital control [145]. The digital signal processing unit generated the control signal for the bias switching operation based on the amplitude of the envelope signal. The dynamic bias switching circuit provided dual supply voltages to the PA’s drain. The dynamic bias switching circuit produced a low supply voltage through the use of a DC–DC converter, whereas the high voltage can be directly sourced from the power supply. The compensation of the gain difference between low- and high-bias voltage conditions was implemented to enhance the linearity of the CMOS PA. The optimization of the threshold voltage for the envelope signal and the low supply voltage level was carried out through analytical means to achieve the highest possible efficiency without trading off its linearity. This was achieved by utilizing the envelope statistics of the LTE signal. This CMOS PA achieved a linear output power of 22 dBm, with a peak PAE of 34.5% at 1.75 GHz. 

B. Kim et al. (2016) proposed a CMOS PA using the reconfigurable adaptive power cell technique. This CMOS PA was configured in a cascode structure that consisted of four differently biased power cells to achieve high linearity [146]. Out of the four power cells, two of them were shut off to operate the CMOS PA at a high frequency. This made the CMOS PA operate at a dual-frequency band. The power cells were designed with different channel widths that give different input average capacitances. Therefore, the total average capacitance variation with regard to the input power may become relatively insignificant as a result of mutual cancelling, which contributes to enhancing its linearity. The maximum output power attained was 27.8 dBm, with a peak PAE of 52.5%. This CMOS PA achieved a linear output power of 17.5 dBm at 650 MHz.

Furthermore, J. Ren et al. (2020) introduced a CMOS PA with an adaptive bias circuit that generated a precise biasing voltage according to the input envelope RF signal [147]. This CMOS PA also employed harmonic termination circuits at the gate of the common-gate PA and the center-tap node of the output transformer. The harmonic termination aided in shorting the second-harmonic components to the ground, which further enhanced the linearity. This CMOS PA achieved a maximum output power of 30.3 dBm, with a peak PAE of 43.1% at 2.4 GHz. The achieved linear output power was 23.5 dBm, with 18.1% linear efficiency.

S. Mariappan et al. (2021) presented a dual-stage CMOS PA with adaptive biasing in which the gate voltage of the driver amplifier was adaptively tuned to be in opposition to the third-order transconductance response compared to the main PA [148]. The biasing voltage was supplied through a current-mirror configured biasing circuit. The opposite third-order transconductances between the stages cancel each other by assisting in mitigating the third-order intermodulation product. This proposed CMOS PA achieved a linear output power of 19 dBm, with a respective PAE of 29% at the 2.45 GHz operating frequency.

A.S. Rawat et al. (2022) presented a CMOS PA with adaptive bias technique for LORA application operating at 919–923 MHz [149]. The driver stage of the CMOS PA utilized a bias voltage modulation mechanism, while the main PA stage used a split bias mechanism to achieve different power level modes. A bias circuit controlled the channel current of the driver and the main PA via the applied gate voltage. The main PA was split into two units, in which each of them was adaptively biased to achieve the optimized linearity performance. A linear output power of 21 dBm, with a respective PAE of 29% at 920 MHz, was obtained. A. S. Rawat et al. (2023) also presented another CMOS PA with a switching mechanism for adaptive-device-sizing biasing [150]. This CMOS PA also employed a high Q-factor compact inductor for efficiency enhancement at linear output power. The adaptive device sizing was implemented to enable the low-power and high-power modes for the PA. The switching of the modes was selected through a NMOS pass transistor. Two individual bias circuits were utilized to conduct the adaptive biasing for the main PA stage. For the low-power mode, only one of the bias circuits was turned ON. In contrast, both bias circuits were turned ON for the high-power mode operation. Thus, the linear output power can be increased with minimum trade-off with efficiency via this method. The proposed CMOS PA achieved a linear output power of 21.5 dBm, with a respective PAE of 22%. Table 3 summarizes the recent state-of-the-art CMOS PAs with linearity enhancement techniques.

Figure 25 depicts the plot of the linear PAE across the achieved linear output power by the aforementioned linearization techniques. It can be observed that most of the employed techniques achieved a linear output power of more than 20 dBm. Most of the CMOS PAs that employ the analog predistortion technique were capable of achieving more than 30% linear PAE. Also, it can be deduced that the CMOS PAs were still limited in delivering a watt-level linear output power, since the PAs were required to operate at a backed-off output power to sustain the linear transmission.

## 5. Bandwidth Enhancement Techniques

A transistor’s gain relatively declines with increasing frequency, with an estimated factor or roll-off slope in the order of 4–6 dB per octave. Thus, in a wideband PA, it is vital to extend the gain response across the required bandwidth with minimum gain deviation. Moreover, the PA’s matching network needs to be designed with a low insertion loss slope for the gain compensation in order to achieve a flat gain as well as high efficiency and linearity. Output matching network is vital in a PA design for realizing optimum output power, efficiency, and linearity. The input and interstage matching networks provide the intended bandwidth extension. Commonly, the input or driver stage of a wideband PA is employed with a negative feedback circuit to minimize the gain loss at higher frequencies [151]. Several bandwidth enhancement techniques were introduced, which include distributed, cascode, feedback, and switched-capacitor PAs.

### 5.1. Distributed Technique

The distributed amplifier (DA) achieves a high bandwidth by subsuming the transistor’s gate-source capacitance (C_gs_) into an artificial transmission line at the input of the PA [152]. The gate and drain line inductors create artificial lines with the C_gs_ and C_ds_ of the amplifiers. Usually, the inductors are replaced with high-impedance microstrip lines in order to achieve low noise and high output power from the PAs. Figure 26 depicts a simplified n-section DA [153].

The signal is amplified via its transconductance and accumulates along the drain line as the signal passes through the PA. The DA method produces a wide bandwidth as a result of the low-pass structures, which produce high cut-off frequencies. The impedance of the input line is defined as
(19)$$Z_{og}={\left(\frac{L_{g}}{C_{ds}}\right)}^{2}$$

In contrast, the impedance of the drain line is defined as
(20)$$Z_{od}={\left(\frac{L_{d}}{C_{ds}}\right)}^{\frac{1}{2}}$$

Since the gate capacitance restricts the achievable bandwidth of the DA, the cut-off frequency is defined as
(21)$$f_{c}=\frac{1}{\pi Z_{og}C_{gs}}$$

Figure 27 shows the frequency response comparison of a DA and PA with lumped elements.

C. Hsiao et al. (2013) proposed a CMOS DA that employs a gate–drain transformer feedback method [154]. This employed feedback method permits the reuse of the traveling signal in order to attain a large gain bandwidth while preserving the low power consumption of the DA. The utilized transformer is miniaturized via a folded transmission lines configuration. A patterned ground shield (PGS) is employed to enhance the quality factor of the transformer. The transformer also has a well-controlled feedback coupling coefficient. This proposed DA achieved a 3 dB bandwidth of 61.3 GHz, with power consumption under 60 mW.

Furthermore, P. Chen et al. (2014) presented a DC-80 GHz compact CMOS DA based on a conventional DA with the gain cell of a cascaded single-stage DA (CSSDA) [155]. To make it area efficient, the artificial transmission lines (ATL) were employed with a micro-strip line instead of a coplanar waveguide. The inter-digital capacitors utilized in this design were optimized for wideband frequency response. The achieved 3 dB bandwidth was 80 GHz, with power consumption under 90 mW.

Y. Zhang et al. (2017) proposed a CMOS DA comprised of two three-stage DAs integrated with an input ATL that served as a preamplifier for broadband impedance matching as well as gain enhancement [156]. In addition, two four-stage DAs were integrated with the output ATLs that served as a medium PA to exhibit the high output power for a wide bandwidth. This proposed DA had a bandwidth operation from 2 to 22 GHz.

M. M. Tarar et al. (2017) presented a wideband stacked distributed CMOS PA [157]. A four-transistor stack was employed to achieve a high output swing and gain. Voltage alignment was achieved by allowing a small AC swing at the gate to be derived from the voltage division between C_gs_ and the external gate capacitance. The four-transistor stack was replicated in the distributed configuration, which contributed to the wideband operation. This PA was capable of operating with a bandwidth from 2 to 16 GHz.

### 5.2. Cascode Technique

A cascode configuration is employed by connecting common-source (CS) and common-gate (CG) transistors in series, as illustrated in Figure 28. The cascode topology is widely utilized in PA designs to enhance the gain and bandwidth [158]. The cascode structure has several advantages, which include high output impedance, high reverse isolation, and less sensitivity to the output resistance of the transistors. High reverse isolation is contributed to by the low Miller feedback capacitance. Moreover, the reduction in Miller capacitance exhibits a 3–4 dB higher gain and increased bandwidth compared to CS structures [159].

In a CS configured device, the reduction in the Miller capacitance in the gate–drain parasitic capacitance (*C_gd_*) generates a frequency pole that makes a small signal gain roll down. The Miller capacitance in the *C_gd_* for a CS amplifier with a voltage gain of *A_v_* = −*g_m_* R_load_ is given by
(22)$$C_{M=}C_{gd}(1+A_{v})$$

The increase in Miller capacitance reduces the bandwidth of the PA. When the cascode transistor is integrated as a load to the drain of the CS amplifier, the Miller effect is eliminated regardless of the load impedance. This feature enhances the bandwidth of the PA. In a two-identical-cascodes device, the small signal open loop voltage gain is defined as [160]
(23)$$A_{vo}=\frac{\nu_{o}}{\nu_{i}}=-g_{m1}r_{o1}x(1+g_{m2}r_{ro2})\approx -g_{m1}r_{o1}g_{m2}r_{o2}\approx -g_{m}^{2}r_{o}^{2}$$
where *g_m_* is the small signal transconductance gain, and *r_o_* is the transistor output impedance. S. Leuschner et al. (2011) proposed a multi-band CMOS PA with a cascode configuration output stage for WCDMA application [161]. A two-stage interstage matching network was utilized to achieve a wideband operation of more than 300 MHz. The cascode PA achieved an efficiency of more than 45% across the operating bandwidth. H. Jeon et al. (2013) proposed a CMOS multi-stage cascode PA with a feedback bias technique [162]. The feedback bias technique was utilized to enhance the linearity and reliability across a wideband frequency range. By utilizing the large parasitic capacitance and low substrate resistivity of the CMOS process, the signal swings were coupled between the ports of the transistors. The leakage signals at the gate of CG structure were employed for the negative feedback. The proposed cascode PA achieved a 3 dB bandwidth of 1.5–2.1 GHz.

H. Wu et al. (2016) proposed a two-stage three-stacked PA with a wideband gain [163]. Wideband load impedance matching was achieved by utilizing modified stacked field-effect transistors with a resistive feedback. To realize the wideband gain frequency response, the compensation between the driver amplifier’s expanding gain response and the last-stage amplifier’s compressing gain response was utilized. The PA achieved a bandwidth of 6.4 GHz from 0.1 to 6.5 GHz. It delivered a maximum output power of 22–24.3 dBm, with a peak efficiency of 13–20%. The achieved efficiency was quite low due to the three-stacked FETs employed at the driver and the last stages, which consume 39.1% and 41% of power, respectively.

In addition, S. Kang et al. (2018) presented a cascode CMOS PA with integrated dynamic body linearizers [164]. The dynamic body linearizers were employed based on the envelope signal injection to the bulk of the CS and CG configured transistors. The dynamic body linearizers mitigated the parasitic capacitances, and these capacitances could limit the modulation bandwidth due to the delays cause by them between the dynamic bias linearizer’s output and the input envelope signals. This cascode PA was capable of performing from 1.7 to 2.0 GHz.

T. Guo et al. (2022) proposed a cascode CMOS PA with a split push–pull configuration at both the driver and main PA stages [165]. This CMOS PA also employed floating-body transistors in both stages. An on-chip bandgap voltage reference was implemented in the design in which it generated the temperature-independent biasing voltage for both the driver and main PA. The design also utilized on-chip transformer baluns for power splitting and combining. The baluns also acted as the wideband impedance matching for the input and output of the CMOS PA. This proposed CMOS PA achieved an operating frequency bandwidth of 4.3–6.4 GHz. This PA achieved a maximum output of 22.2 dBm at 4.9 GHz and a peak PAE of 28% at 4.7 GHz. 

### 5.3. Shunt Feedback Technique

Resistive shunt feedback is a common technique employed to provide wideband matching in PAs. A PA with a resistive feedback provides a good output reflection coefficient. The common practice is to implement a negative resistor feedback from the drain to the gate, as depicted in Figure 29 [166].

Referring to Figure 29, for a PA with a gain of *A_v_*, the input resistance, *R_in_*, is given by
(24)$$R_{in}=\frac{R_{F}}{1+A_{v}}$$

A small *R_F_* provides excellent matching but with the cost of a reduced gain due to the signal feedback through this path. Contrarily, a large *R_F_* provides a high gain but reduces the effect of the feedback.

A straightforward feedback resistor is not adequate when the bandwidth requirements are very large. Thus, some combination of passive or active elements is required in the feedback path. Considering a feedback employed in a CS configured amplifier, a small signal model is shown in Figure 30. Based on Figure 30, the input impedance of a CS configured amplifier is given as
(25)$$Z_{in}^{'}=sL_{bond}+\frac{1}{sC_{gs}}+\frac{g_{m}L_{bond}}{C_{gs}+C_{gd}}$$
(26)$$g_{m,eff}=\frac{g_{m}\nu_{gs}}{\nu_{in}}=\frac{g_{m}}{1+{sg}_{m}L_{bond}}$$

After employing a feedback into the CS amplifier, the input and output impedances and their corresponding gains are defined as
(27)$$Z_{in,o}=\frac{Z_{in}^{'}Z_{f}}{Z_{f}+(A_{v}-1)Z_{in}^{'}}$$
(28)$$Z_{out,o}=\frac{Z_{f}\left(R_{s}∥Z_{in}^{'}\right)}{1+g_{m,eff}\left(R_{s}∥Z_{in}^{'}\right)}$$
(29)$$A_{v}=\frac{R_{L}+g_{m,eff}Z_{f}R_{L}}{R_{L}-Z_{f}}$$

Since $Z_{in}^{'}$ and *g_m_*_,*eff*_ are frequency-dependent parameters, the characteristics of *Z_f_* also correspondingly vary across the frequency band.

S. A. Z. Murad et al. (2010) presented a CMOS PA with a resistive feedback to achieve a wideband flat gain [167]. This PA also adopted a cascode topology with a current reuse method that was utilized to increase the gain at the upper end of the operating band. An inter-stage inductor was employed in between the first and second stages of the PA, which further contributed to the gain flatness. This proposed PA achieved a flat gain with ±0.8 dB deviation from 3.1 to 4.8 GHz and a 3 dB bandwidth of 2.8–5.0 GHz.

S. A. Z. Murad et al. (2010) also presented a wideband CMOS PA, which achieved a flat gain by employing a shunt–shunt feedback [168]. This PA utilized a current reuse technique in the first stage to enhance the gain at the upper band of the operating frequency. The second stage of the PA consisted of a shunt and series peaking inductors with a resistive feedback and a shunt–shunt feedback. The resonance circuit in this PA provided a narrow band characteristic that increased the gain at the upper end of the operating frequency (7 GHz). A shunt inductor was inductively peaking at the low frequency region (3 GHz), thus realizing a wide-bandwidth gain. A series peaking inductor was employed between the first and second stages in order to increase the middle band’s gain. This contributed to the gain flatness across the wide bandwidth. The employed resistive feedback aided in further flattening the gain. A shunt–shunt feedback using a resistor and a capacitor further increased the bandwidth and improved the wideband output matching. This PA achieved a flat gain with ±0.5 dB deviation from 3.0 to 7.0 GHz as well as a 3 dB bandwidth of 2.8–7.4 GHz.

In addition, N. Ginzberg et al. (2023) presented a wideband CMOS PA that implemented a cascode topology with a resistive shunt–shunt feedback [169]. This CMOS PA was biased in the class B operation region. The design adopted simultaneous input and output matching via the feedback resistor. The biasing network was also resistively implemented in order to achieve a wideband response. The shunt–shunt resistive feedback technique was required to conduct impedance matching as well as stability. In order to achieve both unconditional stability and optimum input matching, a 15 Ω resistor was employed at the input. This proposed CMOS PA achieved an operating frequency bandwidth of 0.33–2.5 GHz. The maximum output power delivered was 21.5 dBm, with a peak PAE of 52.4% at 1 GHz.

### 5.4. Switched Capacitor Technique

A switched capacitor PA (SCPA) has been extensively trending in recent years, and it employs an array of capacitors that are tuned between the supply and ground [170,171,172,173,174,175,176,177]. Figure 31 shows the schematic of the basic switched capacitor (SC) architecture [178].

The amplitude of the signal is created by a capacitive voltage division between the switched-on capacitors and switched-off capacitors as follows [179]:(30)$$Vout=\frac{C_{ON}}{C_{ON}+C_{OFF}}\cdot V_{DD}=\frac{C_{ON}}{C}\cdot V_{DD}$$
where *C_ON_* is the equivalent capacitance of the switched-on capacitors, *C_OFF_* is the equivalent capacitance of the switched-off capacitors, and *C* is the total array capacitance. The capacitor array is integrated with a band-pass matching network that resonates the total capacitance at the frequency of application.

The output power is obtained by considering the output voltage as a fundamental component of the Fourier-series expansion of a square wave signal. The output voltage and power are given as
(31)$$Vout=\left(\frac{2}{\pi }\right)\left(\frac{n}{N}\right)V_{DD}$$
(32)$$P_{out}=\left(\frac{2}{\pi^{2}}\right){\left(\frac{n}{N}\right)}^{2}\frac{V_{DD}^{2}}{R_{opt}}$$
where *n* is the number of switched-on capacitors, and *N* is the total capacitance. The PAE of the SCPA is given by
(33)$$PAE=\frac{P_{out}}{P_{DC,SCPA}+P_{SW,in}}$$
where *P_SW_*_,*in*_ is the dynamic losses of the SCPA core drivers and the clock distribution network, and *P_DC_*_,*SCPA*_ is the average power consumption of the PA core. *P_DC_*_,*SCPA*_ also takes into account the ohmic losses of the switch resistances as well as the dynamic power required for the charging and discharging of the capacitor array [180].

S. Goswami et al. (2014) presented a CMOS PA with a frequency agile solution that was tunable over a wide-bandwidth frequency [181]. This CMOS PA utilized a class D configured out-phasing technique with an integrated tunable matching network. The tunable matching network was realized via a series combining a transformer and a digitally tuned capacitor bank. The unit capacitance was employed in unary weighted mode. The proposed PA was implemented using 45 nm CMOS SOI technology with a supply headroom of 1.8 V. It was capable of operating from 1.44 to 3.41 GHz, with a bandwidth of 1.97 GHz. This PA delivered a maximum output power of 27–28.2 dBm across the frequency, with a peak efficiency of 25–30%. The achieved linear output power was 21.6–23.4 dBm, with an efficiency of 9.8–17.4% across the frequency of operation.

S. Yoo et al. (2011) presented an SCPA for EER or polar transmitters without the need for a supply modulator [182]. A baseband DSP was employed to generate a polar-modulated signal, and the number of capacitors to be switched were controlled by the digital envelope input signal to a thermometer decoder. This SCPA can be considered as a capacitive power combination of multiple switching PAs in which the output power is the combination of the charge redistribution on the capacitors. This proposed PA was capable of operating from 1.8 to 2.8 GHz.

Furthermore, W. Yuan et al. (2016) proposed a digital-based class G quadrature SCPA, which combined the I and Q signals on a shared capacitor array [183]. This proposed Q-SCPA did not need a wideband phase modulator or a delay matching circuitry, unlike its polar or EER counterparts. The capacitor array was subdivided in such a way that half of the array was located in the path of the individual 1/Q paths. The subarrays were also divided into unit capacitors. The unit capacitors had individual driver chains that could be switched on and off. A 3 dB bandwidth of 400 MHz from 1.9 to 2.3 GHz was achieved by the proposed SCPA.

Z. Bai et al. (2018) proposed a multiphase SCPA consisting of C-2C split array capacitors [184]. This SCPA was designed for 16-bit resolution for additional linearization or calibration states using DPD. C-2C arrays are extensively utilized due to the low ratio of the maximum capacitance to the minimum capacitance, which offers a better impedance matching [185,186]. However, this comes with the expense of increased sensitivity to parasitic effects [187,188]. The C-2C method consumes less area compared to binary weight arrays because C-2C arrays linearly increase with the resolution. The proposed SCPA achieved a 3 dB bandwidth of 700 MHz from a center frequency of 1.8 GHz. Moreover, Z. Bai et al. (2019) also presented a frequency-tunable SCPA that achieved an operating bandwidth of more than 1 GHz [189]. This SCPA was employed in series with an inductor and a digitally programmable capacitor (DPC). The frequency tuning of the SCPA resonance was conducted by the DPC. The SCPA comprised an 8-bit array that consisted of 4-unary and 4-binary weighted bits. The switch and driver stages were employed as custom cells that can be tiled with nearby cells in order to mitigate parasitic capacitance. This proposed SCPA was capable of tuning its frequency from 1.4 to 2.5 GHz of bandwidth, while maintaining an output power of 20–21.7 dBm.

I. Raja et al. (2020) proposed a reconfigurable RF transmitter with an integrated tunable class E PA [190]. A DSP was utilized to minimize the sampling spurs up to the harmonics of the carrier, which avoided the requirement for sharp RF filters. Other than that, a DPD was also employed as a linearizer for the class E PA. Clock correction circuitry was implemented for rectifying the duty cycle and quadrature errors. In the impedance matching network, the capacitors were made tunable to provide wave shaping across the frequency range, while using a single, fixed-series inductor. The switching transistors were the control bits for the tunable capacitor bank. The capacitors were segmented into smaller capacitors in order to achieve an effective tuning. This proposed PA had a bandwidth of 1.75 GHz, from 0.75 to 2.5 GHz. This PA delivered a maximum linear output power of 6.5–13.0 dBm, with a linear efficiency of 5–27%, when tested with a 25-kS/s 16-QAM signal. In addition, the finite impulse response (FIR) filters, DPD segments, and the RF choke inductor were employed off-chip in this design. Table 4 summarizes the recent state-of-the-art CMOS PAs with bandwidth enhancement techniques.

Figure 32 shows the summary of the frequency bandwidths achieved by the proposed bandwidth enhancement techniques. Based on Figure 32, the largest bandwidth is achieved by the distributed technique, which has a range from the DC level to 80 GHz. However, the other performance parameters of the PA such as its output power and PAE are very low for this technique. The other techniques are capable of achieving an operating bandwidth of more than 1 GHz, with an optimum output power and PAE for its given application. The switched-capacitor-based CMOS PA has the most flexibility and configurability in tuning its parameters according to its need, since its impedance can be varied.

## 6. Conclusions

This review paper discusses the trending state-of-the-art CMOS RFPAs in wireless communication systems. As the number of wireless devices and mobile data usage increase, there is a growing need for enhancements and upgrades to current wireless communication systems. This review provides a concise discussion of the performance metrics that are important for designing a CMOS PA, followed by an overview of recent research on CMOS PAs’ performance enhancement techniques, including efficiency, linearity, and bandwidth. This review concludes that there is a trade-off between efficiency, linearity, and bandwidth in CMOS PAs, and several enhancement techniques are developed to improve their performance. The on-chip transformer and out-phasing techniques are capable of delivering watt-level maximum output power for CMOS PAs. This is due to the usage of power-combining architectures in these techniques that boost the maximum output power. It can also be deduced that CMOS PAs are still limited in delivering watt-level linear output power, since PAs are required to operate at backed-off output power to sustain the linear transmission. The switched-capacitor-based CMOS PA has the most flexibility and configurability in tuning its parameters according to its need, since its impedance can be tuned.

## Figures and Tables

**Figure 1 micromachines-14-01551-f001:**
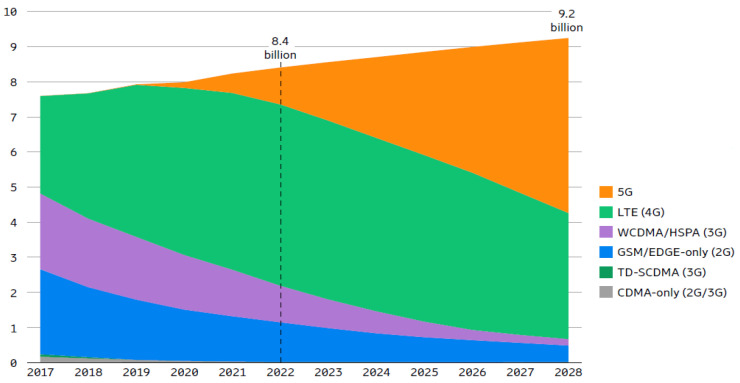
Mobile subscription trend [1].

**Figure 2 micromachines-14-01551-f002:**
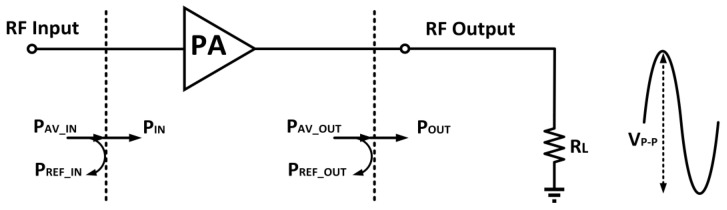
Depiction of an output power definition in a PA.

**Figure 3 micromachines-14-01551-f003:**
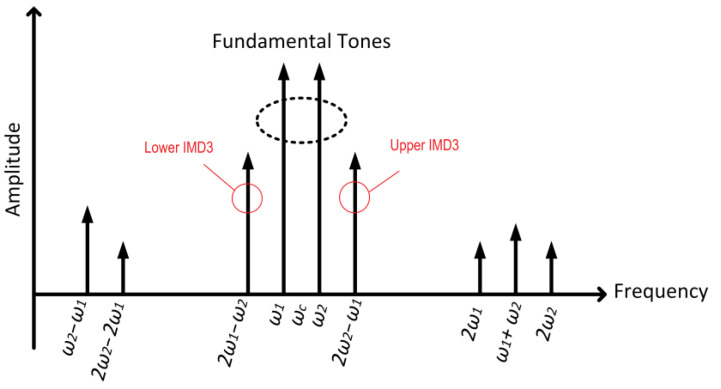
The IMD components induced by a PA.

**Figure 4 micromachines-14-01551-f004:**
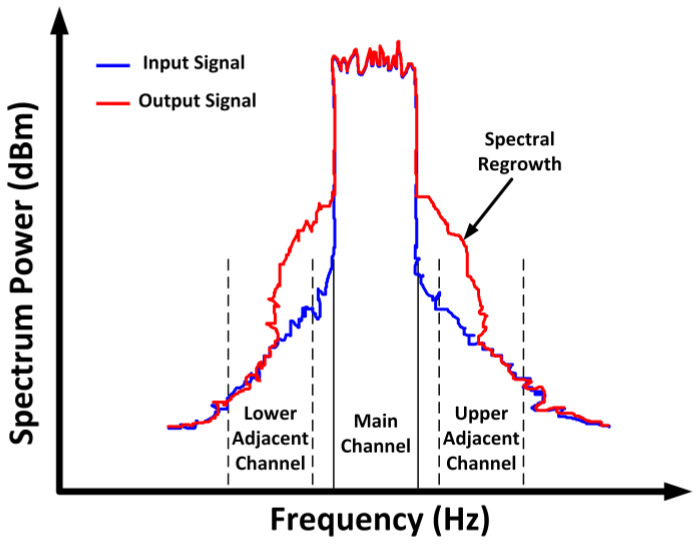
Spectral regrowth of a PA under a modulated signal.

**Figure 5 micromachines-14-01551-f005:**
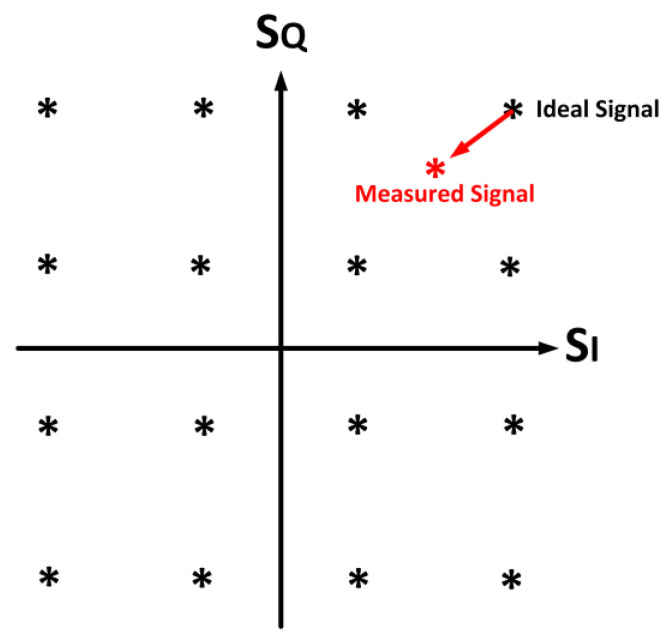
EVM constellation diagram.

**Figure 6 micromachines-14-01551-f006:**
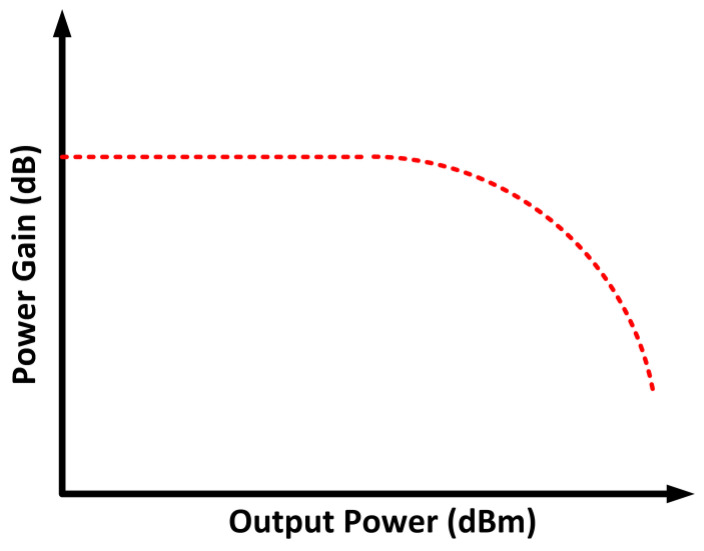
AM–AM feature of a PA across the output power.

**Figure 7 micromachines-14-01551-f007:**
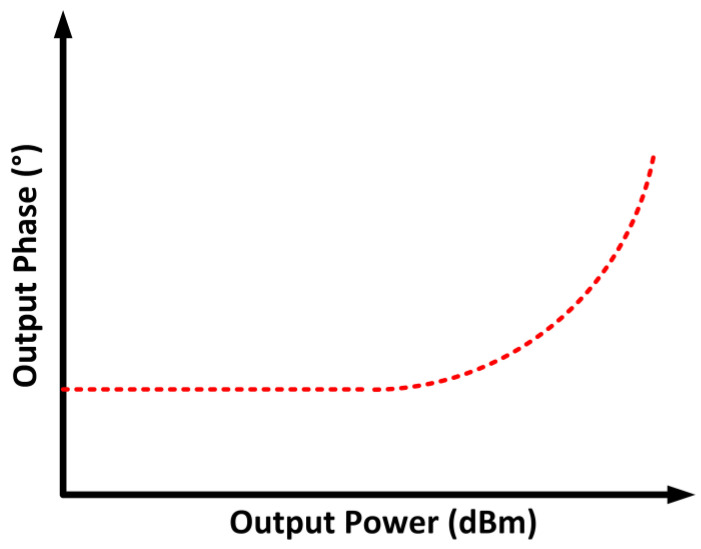
AM–PM feature of a PA across the output power.

**Figure 8 micromachines-14-01551-f008:**
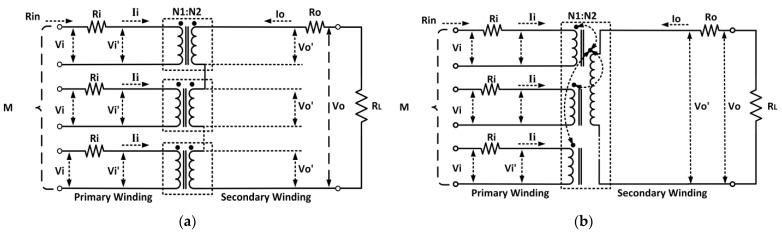
(**a**) Series transformer combination structure. (**b**) Parallel transformer combination structure.

**Figure 9 micromachines-14-01551-f009:**
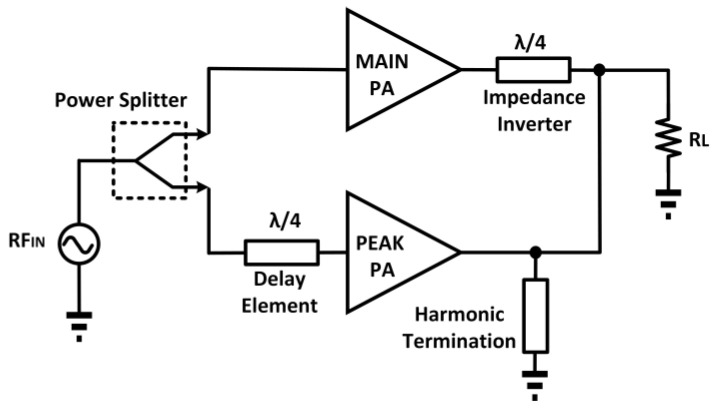
Block diagram of a classic DPA.

**Figure 10 micromachines-14-01551-f010:**
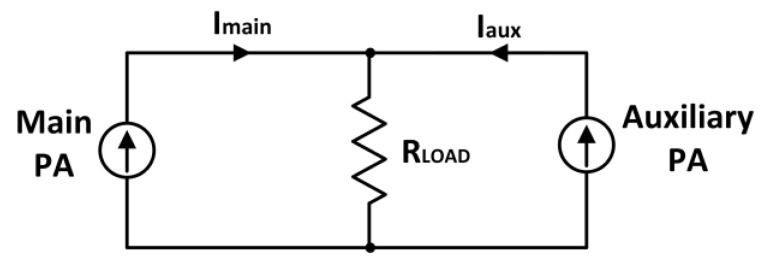
The concept of Doherty PA.

**Figure 11 micromachines-14-01551-f011:**
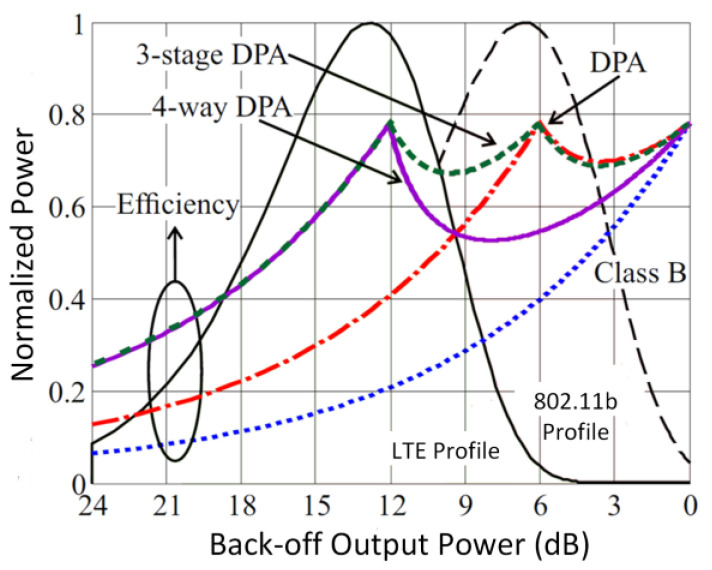
The efficiency profile of DPA.

**Figure 12 micromachines-14-01551-f012:**
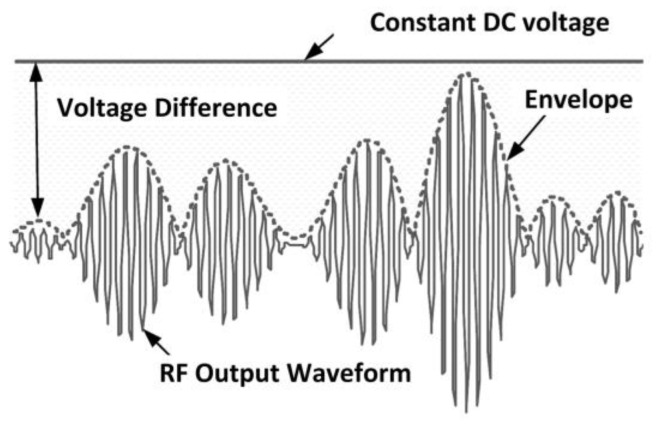
Voltage difference between supply voltage and modulated RF output signal waveform.

**Figure 13 micromachines-14-01551-f013:**
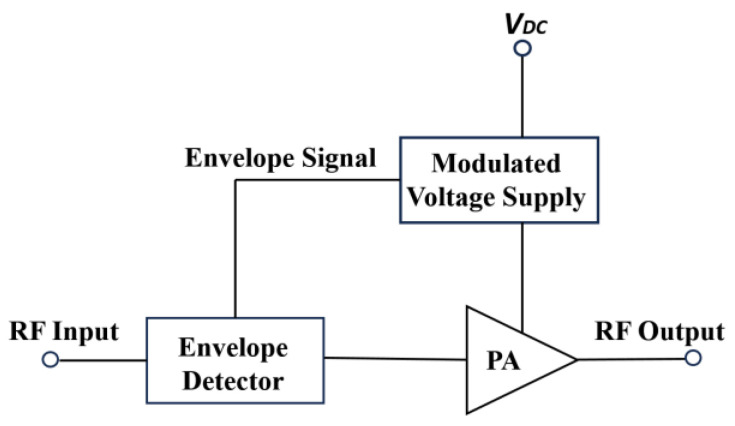
Block diagram of the ETPA.

**Figure 14 micromachines-14-01551-f014:**
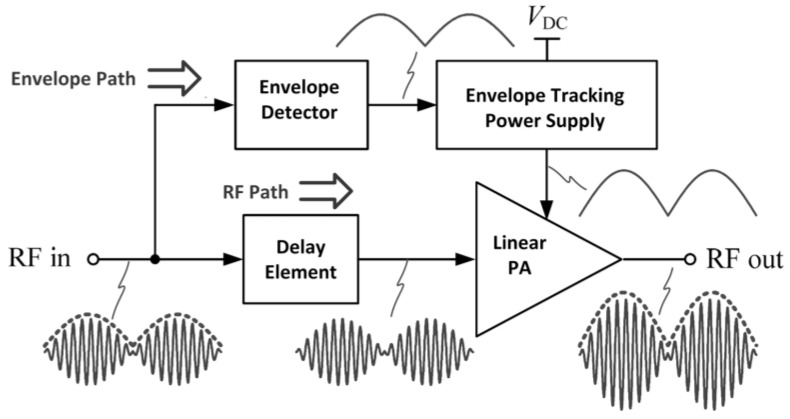
Key waveforms in an ETPA.

**Figure 15 micromachines-14-01551-f015:**
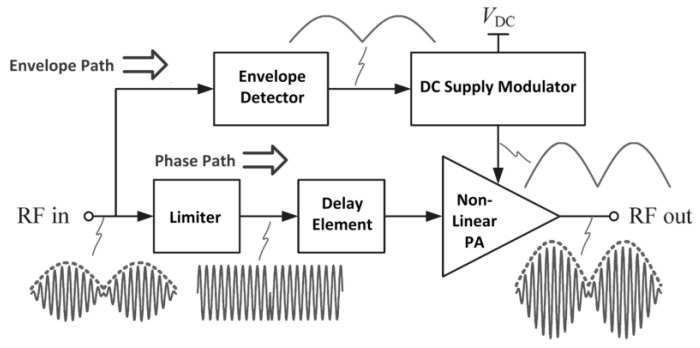
Key waveforms in an EERPA.

**Figure 16 micromachines-14-01551-f016:**
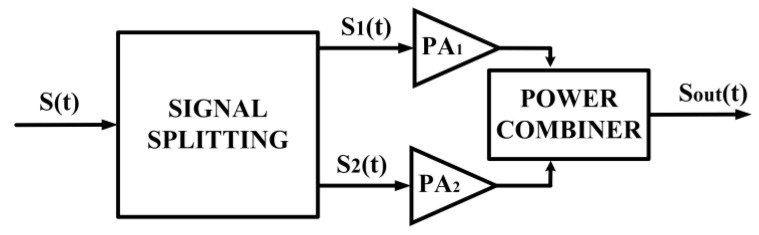
Architecture block diagram of out-phasing.

**Figure 17 micromachines-14-01551-f017:**
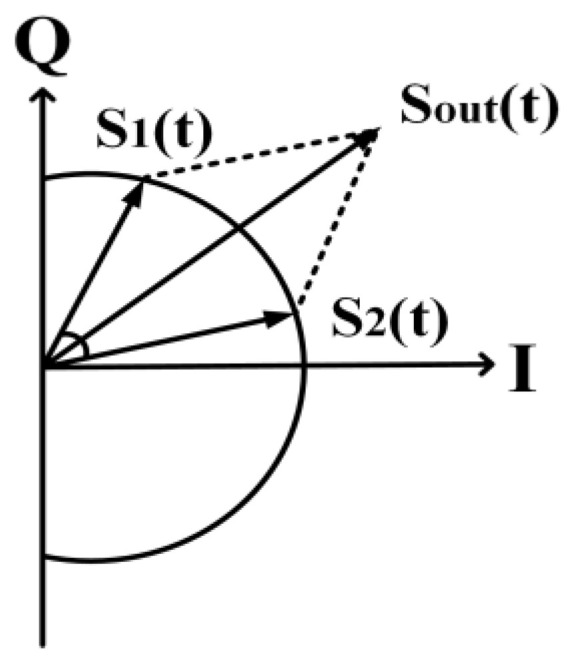
Operation principle of out-phasing.

**Figure 18 micromachines-14-01551-f018:**
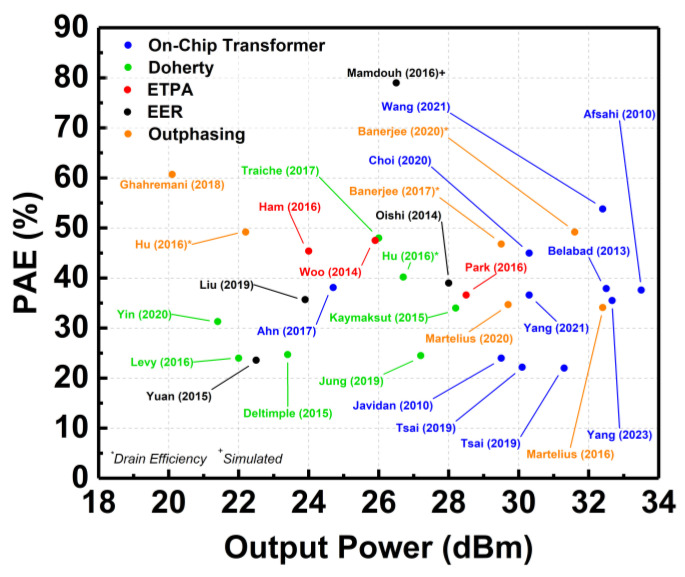
PAE across output power for the proposed efficiency enhancement techniques [55,58,60,61,62,63,64,65,66,67,72,73,74,77,78,79,80,85,86,87,89,90,91,92,99,100,101,102,103,104].

**Figure 19 micromachines-14-01551-f019:**
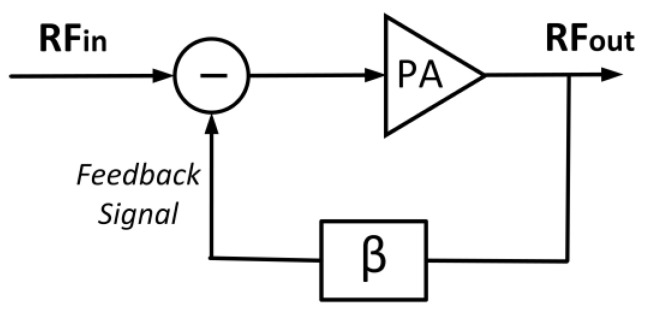
Principle of feedback technique.

**Figure 20 micromachines-14-01551-f020:**
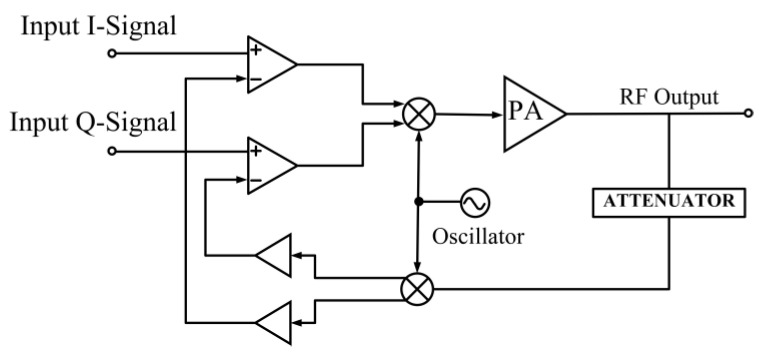
Cartesian feedback technique.

**Figure 21 micromachines-14-01551-f021:**
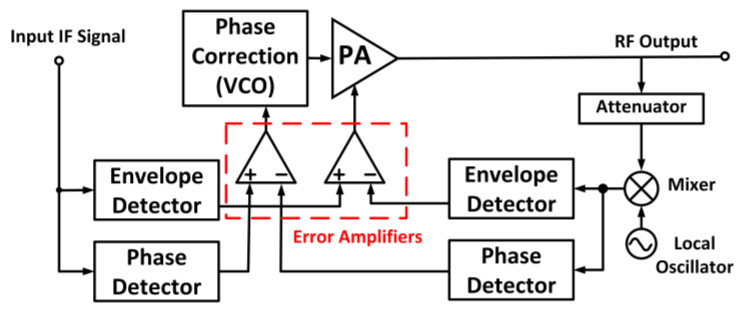
Block diagram of a polar feedback system.

**Figure 22 micromachines-14-01551-f022:**
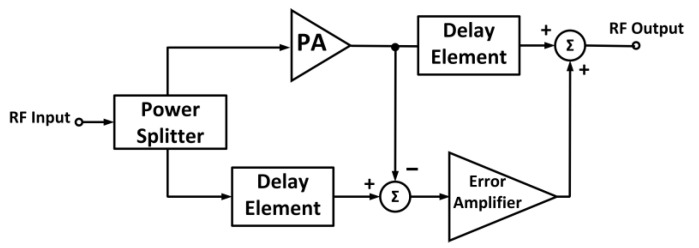
Block diagram of the feed-forward-based PA.

**Figure 23 micromachines-14-01551-f023:**
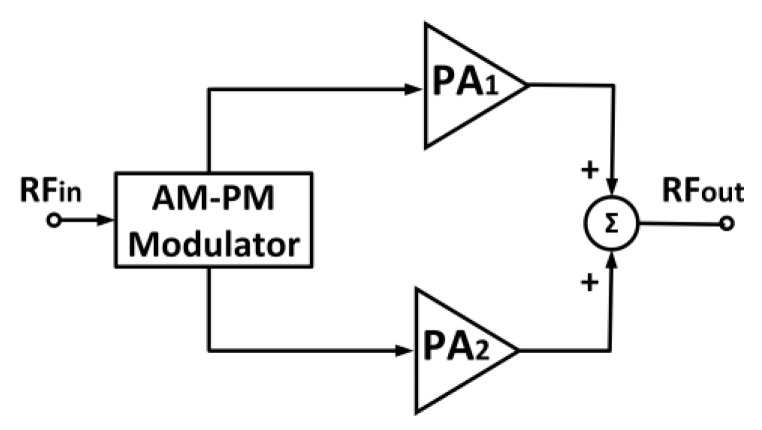
Block diagram of LINC technique.

**Figure 24 micromachines-14-01551-f024:**
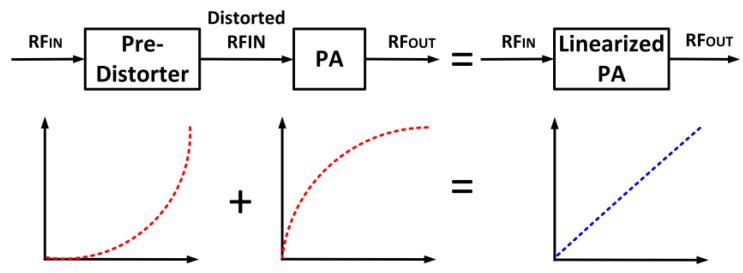
The concept of pre-distortion.

**Figure 25 micromachines-14-01551-f025:**
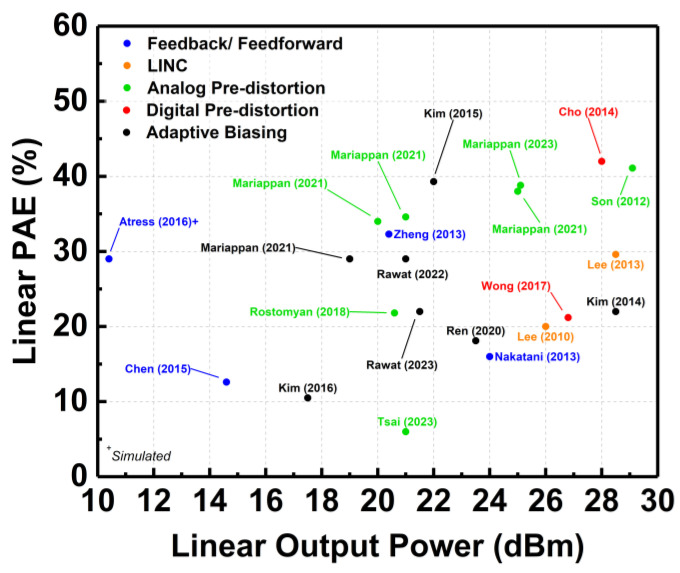
Linear PAE across linear output power for the proposed linearity enhancement techniques [116,118,119,121,123,124,129,132,133,134,135,136,137,142,143,144,145,146,147,148,149,150].

**Figure 26 micromachines-14-01551-f026:**
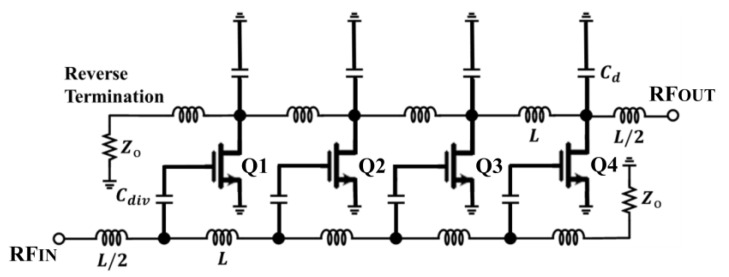
Simplified n-section DA.

**Figure 27 micromachines-14-01551-f027:**
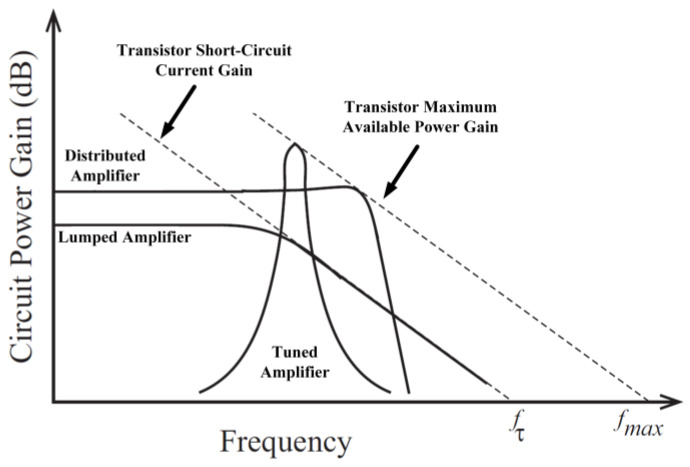
Frequency response comparison.

**Figure 28 micromachines-14-01551-f028:**
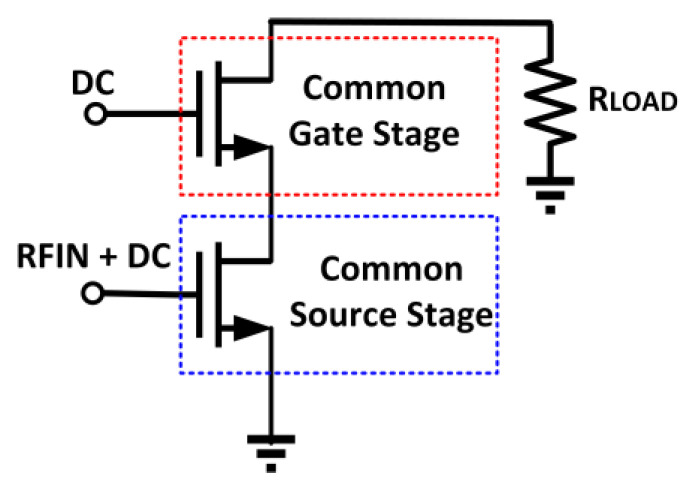
Cascode PA configuration.

**Figure 29 micromachines-14-01551-f029:**
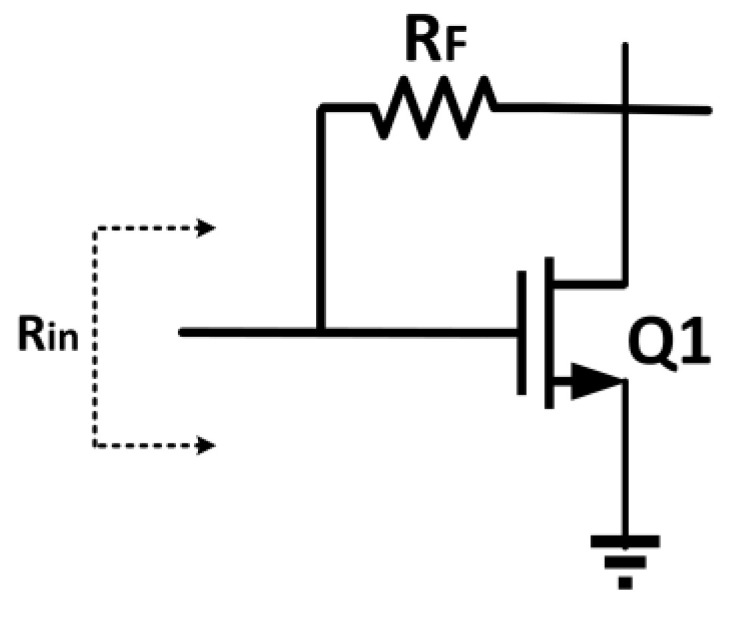
Configuration of a resistive feedback.

**Figure 30 micromachines-14-01551-f030:**
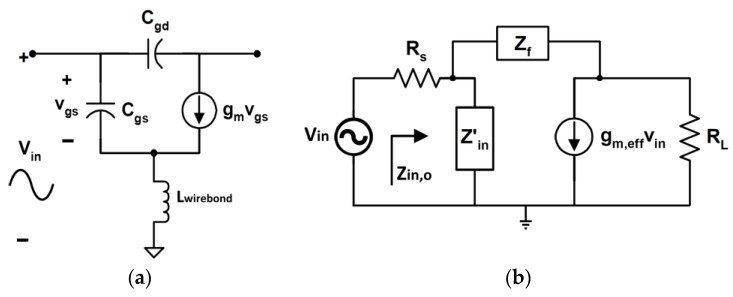
(**a**) Small signal model of CS; (**b**) small signal model of CS with feedback.

**Figure 31 micromachines-14-01551-f031:**
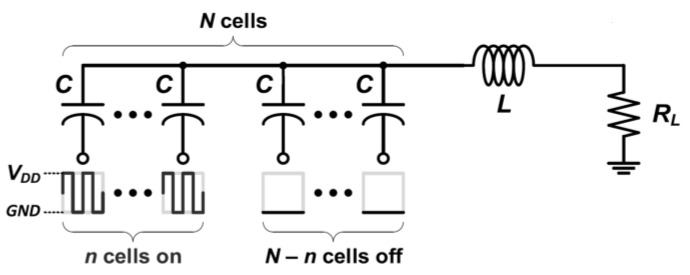
Schematic of the basic switched capacitor (SC) architecture.

**Figure 32 micromachines-14-01551-f032:**
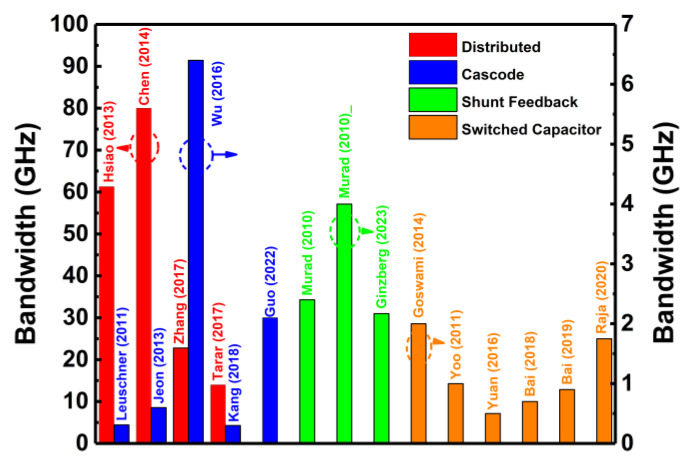
Frequency bandwidth achieved by the proposed bandwidth enhancement techniques [154,155,156,157,161,162,163,164,165,167,168,169,181,182,183,184,189,190].

**Table 1 micromachines-14-01551-t001:** IMD components.

Order	Terms	*a* _1_ *v* ^3^	*a* _2_ *v* ^2^	*a* _3_ *v* ^3^
0	0		1	
1st	*ω* _1_	1		9/4
	*ω* _2_	1		9/4
2nd	2*ω*_1_		1/2	
	2*ω*_2_		1/2	
	*ω*_1_ ± *ω*_2_		1	
3rd	3*ω*_1_			1/4
	3*ω*_2_			1/4
	2*ω*_1_ ± *ω*_2_			3/4
	2*ω*_2_ ± *ω*_1_			3/4

**Table 2 micromachines-14-01551-t002:** Summary of the recent state-of-the-art CMOS PAs with efficiency enhancement techniques.

Architecture	Year	Tech (nm)	V_DD_ (V)	Freq. (GHz)	Max. P_out_ (dBm)	Peak PAE (%)	Ref.
On-Chip Transformer	2010	180	3.3	0.9	29.5	24.0	[55]
	2010	65	3.3	2.4	33.5	37.6	[58]
	2013	180	3.3	2.4	32.5	37.9	[60]
	2017	40	2.8	1.75	24.7	38.1	[61]
	2019	180	5.0	5.2	30.1	22.2	[62]
	2019	180	3.3	5.3	31.3	22.0	[63]
	2020	180	4.6	0.82–1.0	30.3	45.0	[64]
	2021	40	2.5	1.2–2.8	32.4	53.8	[65]
	2021	40	2.4	2.4	30.3	36.6	[66]
	2023	40	2.5	2.0	32.67	35.5	[67]
Doherty	2015	65	2.5	2.5	23.4	24.7	[72]
	2016	65	3.0	3.71	26.7	40.2 *	[73]
	2015	40	1.5	1.7–2.1	28.2	34.0	[74]
	2016	45	2.4	14.0	22.0	24.0	[77]
	2017	130	2.6	3.0	26.0	48.0	[78]
	2019	55	5.5	5.8	27.2	24.5	[79]
	2020	40	1.1	1.5	21.4	31.3	[80]
ETPA	2014	320	3.4	0.837	25.9	47.5	[85]
	2016	180	4.7	1.7	28.5	36.6	[86]
	2016	180	3.3	0.78	24.0	45.4	[87]
EER	2020	130	3.0	1.0	26.5	79	[89] ^+^
	2014	90	3.3	1.95	28.0	39	[90]
	2015	130	2.5	2.4	22.5	23.6	[91]
	2019	65	2.4	2.4	23.9	35.7	[92]
Out-phasing	2016	40	1.2	5.9	22.2	49.2 *	[99]
	2016	28	3.6	1.8	32.4	34.1	[100]
	2017	45	2.4	2.4	29.5	46.8 *	[101]
	2018	65	1.25	1.8	20.1	60.7	[102]
	2020	28	3.6	1.7	29.7	34.7	[103]
	2020	45	2.4	2.4	31.6	49.2 *	[104]

* drain efficiency; ^+^ simulated.

**Table 3 micromachines-14-01551-t003:** Summary of the recent state-of-the-art CMOS PAs with linearity enhancement techniques.

Architecture	Year	Tech (nm)	V_DD_ (V)	Freq. (GHz)	Linear. P_out_ (dBm)	Linear PAE (%)	Ref.
Feedback/Feedforward	2016	130	1.2	0.868	10.4	29.0	[116] ^+^
	2013	150	3.3	1.75	24.0	16.0	[118]
	2013	65	2.0	2.0	20.4	32.3	[119]
	2015	180	3.6	24.0	14.6	12.6	[121]
LINC	2010	180	3.3	1.95	26.0	20.0	[123]
	2013	130	3.5	1.95	28.5	29.6	[124]
Pre-Distortion	2012	180	3.5	1.95	29.1	41.1	[129]
	2018	45	4.8	15.0	20.6	21.8	[132]
	2021	180	3.3	0.8–3.3	20.0	34.0	[133]
	2021	180	3.3	0.4–2.8	21.0	34.6	[134]
	2021	180	3.3	1.7–2.7	25.0	38.0	[135]
	2023	180	3.3	1.7–2.7	25.1	38.8	[136]
	2023	180	3.3	5.3	21.0	6.0	[137]
	2014	180	3.5	1.7–2.0	28.0	42.0	[142]
	2017	28	3.3	2.4	26.8	21.2	[143]
Adaptive Biasing	2014	180	3.5	0.87	28.5	22.0	[144]
	2015	180	3.3	1.75	22.0	39.3	[145]
	2016	40	-	0.65	17.5	10.5	[146]
	2020	130	2.5	2.4	23.5	18.1	[147]
	2021	180	3.3	2.45	19.0	29.0	[148]
	2022	180	3.3	0.92	21.0	29.0	[149]
	2023	130	3.3	0.92	21.5	22.0	[150]

^+^ simulated.

**Table 4 micromachines-14-01551-t004:** Summary of the recent state-of-the-art CMOS PAs with bandwidth enhancement techniques.

Architecture	Year	Tech (nm)	V_DD_ (V)	Freq. (GHz)	Bandwidth (GHz)	Ref.
Distributed	2013	90	2.2	DC–61.3	61.3	[154]
	2014	40	1.7	DC–80.0	80.0	[155]
	2017	180	2.8	1.0–23.8	22.8	[156]
	2017	130	3.5	2.0–16.0	14.0	[157]
Cascode	2011	65	3.4	1.64–1.95	0.31	[161]
	2013	180	3.4	1.5–2.1	0.6	[162]
	2016	180	6.0	0.1–6.5	6.4	[163]
	2018	180	3.3	1.7–2.0	0.3	[164]
	2022	130	3.3	4.3–6.4	2.1	[165]
Shunt Feedback	2010	180	1.8	2.6–5.0	2.4	[167]
	2010	180	1.8	3.0–7.0	4.0	[168]
	2023	65	3.3	0.33–2.5	2.17	[169]
Adaptive Biasing	2014	45	2.0	1.3–3.3	2.0	[181]
	2011	90	2.5	1.8–2.8	1.0	[182]
	2016	65	2.4	1.8–2.3	0.5	[183]
	2018	65	2.4	1.4–2.1	0.7	[184]
	2019	65	2.4	1.4–2.5	0.9	[189]
	2020	130	1.5	0.75–2.5	1.75	[190]

## Data Availability

Not applicable.

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
