# Peer review of "A State-of-the-Art Review on CMOS Radio Frequency Power Amplifiers for Wireless Communication Systems"

_micromachines, 2023, doi:10.3390/mi14081551_

Round 1

Reviewer 1 Report

This review paper discusses the trending state-of-the-art CMOS RFPAs in wireless communication systems. This review provides a concise discussion of the performance metrics that are important for designing a CMOS PA, followed by an overview of recent research on CMOS PAs performance enhancement techniques, including efficiency, linearity, and bandwidth. Based on these works, this paper concludes that a trade-off between efficiency, linearity, and bandwidth in CMOS PAs is desired. Further, several enhancement techniques are introduced to improve related performance. From my point of view, the review is very comprehensive and enlightening. It deserves to be accepted by the journal.

Some minor modifications are suggested as follows:

1. Some figures such as Fig. 48, and Fig. 16 in the paper are not clear, please replace them with high-quality ones.

2. If possible, please add some of your own opinions to the review.

Author Response

Thank you very much for the comments. All the comments have been addressed and revised.

Reviewer 2 Report

This paper provides a comprehensive review of CMOS PAs, but it is unfortunate that it does not address the recent reports on mm-wave PAs, which have become one of the most important issues in recent years.

In the section on BW enhancement techniques, some wideband PAs covering the mm-wave range are introduced, but the information on linearity and efficiency is all below 10 GHz. This inconsistency makes the scope of the paper unclear.

I am not sure if transformers should be considered from the perspective of efficiency improvement. Transformers are mainly used for power combining, so it is questionable whether they should be discussed in the same category as Doherty, ET, and EER.

.

There are many duplications of figures. The figures in the following lines of are duplicated.

           167

           180

           202 & 203

           206 & 207

           467 & 470

           505 & 506

           560 & 561

           782 & 783

           818 & 819

           850 & 851

           1105 & 1106

           1119 & 1120

           1160 & 1161

           1229 & 1230

           1409 & 1412

Line #161 It would be good to provide the ACPR values required for communication systems.

Line #450 It would be good to add information on the practical communication techniques that use the linearity enhancement techniques explained.

Line #952 I think the comparison between APD and DPD is insufficient. It seems that DPD is more widely used, but it is unfortunate that there are fewer contents than APD and it is excluded from the comparison table.

Line #1243-1246 The content of lines 1238-1241 is the same.

There are a few grammatical errors found. Proofreading is required.

For example, "The linear amplification using non-linear component (LINC) is usually employ with  in #810"

Author Response

(The authors gave the same response as above.)
